# Disulfide bond in SUN2 regulates dynamic remodeling of LINC complexes at the nuclear envelope

Rahul Sharma ⓘ, Martin W Hetzer ⓘ

The LINC complex tethers the cell nucleus to the cytoskeleton to regulate mechanical forces during cell migration, differentiation, and various diseases. The function of LINC complexes relies on the interaction between highly conserved SUN and KASH proteins that form higher-order assemblies capable of load bearing. These structural details have emerged from in vitro assembled LINC complexes; however, the principles of in vivo assembly remain obscure. Here, we report a conformation-specific SUN2 antibody as a tool to visualize LINC complex dynamics in situ. Using imaging, biochemical, and cellular methods, we find that conserved cysteines in SUN2 undergo KASH-dependent inter- and intra-molecular disulfide bond rearrangements. Disruption of the SUN2 terminal disulfide bond compromises SUN2 localization, turnover, LINC complex assembly in addition to cytoskeletal organization and cell migration. Moreover, using pharmacological and genetic perturbations, we identify components of the ER lumen as SUN2 cysteines redox state regulators. Overall, we provide evidence for SUN2 disulfide bond rearrangement as a physiologically relevant structural modification that regulates LINC complex functions.

## Introduction

Apart from harboring and organizing the genome, the cell nucleus plays a crucial role in integrating intra and extracellular mechanical cues to generate an appropriate cellular response. The linker of cytoskeleton and nucleoskeleton (LINC) is a specialized protein complex that forms the molecular basis of this mechanical signaling by physically connecting the cytoskeleton to the nuclear lamina (NL). The LINC complex is comprised of a highly conserved Sad1/UNC-84 (SUN) domain containing protein and Klarsicht/ANC-1/ Syne-1 homology (KASH) domain containing proteins that traverse the inner and outer nuclear membranes (INM/ONM), respectively. SUN and KASH proteins directly interact in the nuclear envelope (NE) lumen (i.e., perinuclear space), which is continuous with the ER lumen. In addition, SUN and KASH proteins interact with the NL and

Actin/microtubule network to physically tether the nucleus to the rest of the cell (Crisp et al, 2006) (Fig 1A). The LINC complex is important for key cellular processes like cell migration, nuclear positioning and anchorage, cell differentiation, and mechanotransduction (Zhang et al, 2007, 2009; Gant Luxton et al, 2010; Lombardi et al, 2011; Cain et al, 2018; Déjardin et al, 2020; Carley et al, 2021). Moreover, because SUN and KASH proteins are associated with muscular dystrophy, cardiomyopathy, and cancer (Chen et al, 2012; Meinke et al, 2014; Matsumoto et al, 2015; Maurer & Lammerding, 2019; Stewart et al, 2019; Battey et al, 2020), the understanding of the molecular basis of LINC complex assembly and function could lead to future therapeutics.

Mammals express five different SUN-containing proteins of which SUN1 and SUN2 are ubiquitous and partially redundant (Lei et al, 2009). Apart from the conserved SUN domain, SUN proteins contain additional functional domains: (i) a lamin-binding domain that interacts with A/B-type lamins and helps to anchor the protein in the INM (Crisp et al, 2006), (ii) two coiled coil domains (CC1, CC2) that reside in the NE lumen and seem to regulate protein conformation (Nie et al, 2016). In addition, five different KASH-containing proteins (NESPRIN/SYNE) and multiple isoforms have been reported in mammals generating a remarkable diversity among the KASH family (Zhang et al, 2005). NESPRINS are megadalton proteins comprising of cytoskeleton-interacting domain, spectrin repeat region, and a small 20–30 AA KASH domain (Meinke & Schirmer, 2015) (Fig 1A). Although in vitro data suggest promiscuous interaction among different SUN and NESPRIN proteins (Kim et al, 2015), there is evidence for specialized SUN:NESPRIN pairs. For example, SUN1:NESPRIN1 and SUN2:NESPRIN2 pairs are required for forward and rearward nuclear movement in migrating cells, respectively (Zhu et al, 2017). How endogenous SUN proteins choose NESPRIN partners remains unclear. Moreover, acquisition of new forms of SUN and NESPRIN proteins in vertebrates points towards an underexplored functional diversity and emergence of LINC-independent functions over the course of evolution. Currently, there are limited tools to study SUN oligomerization and SUN–KASH protein interaction in vivo (Hennen et al, 2018).

X-ray crystallography of in vitro assembled truncated fragments of SUN and KASH proteins have been invaluable in providing insights into SUN–KASH interaction and LINC complex assembly. The

Molecular and Cell Biology Laboratory, Salk Institute for Biological Studies, La Jolla, CA, USA

Correspondence: martin.hetzer@ist.ac.at

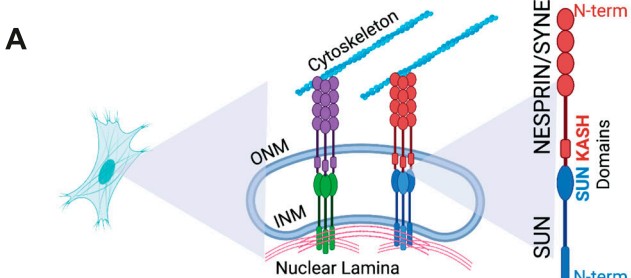

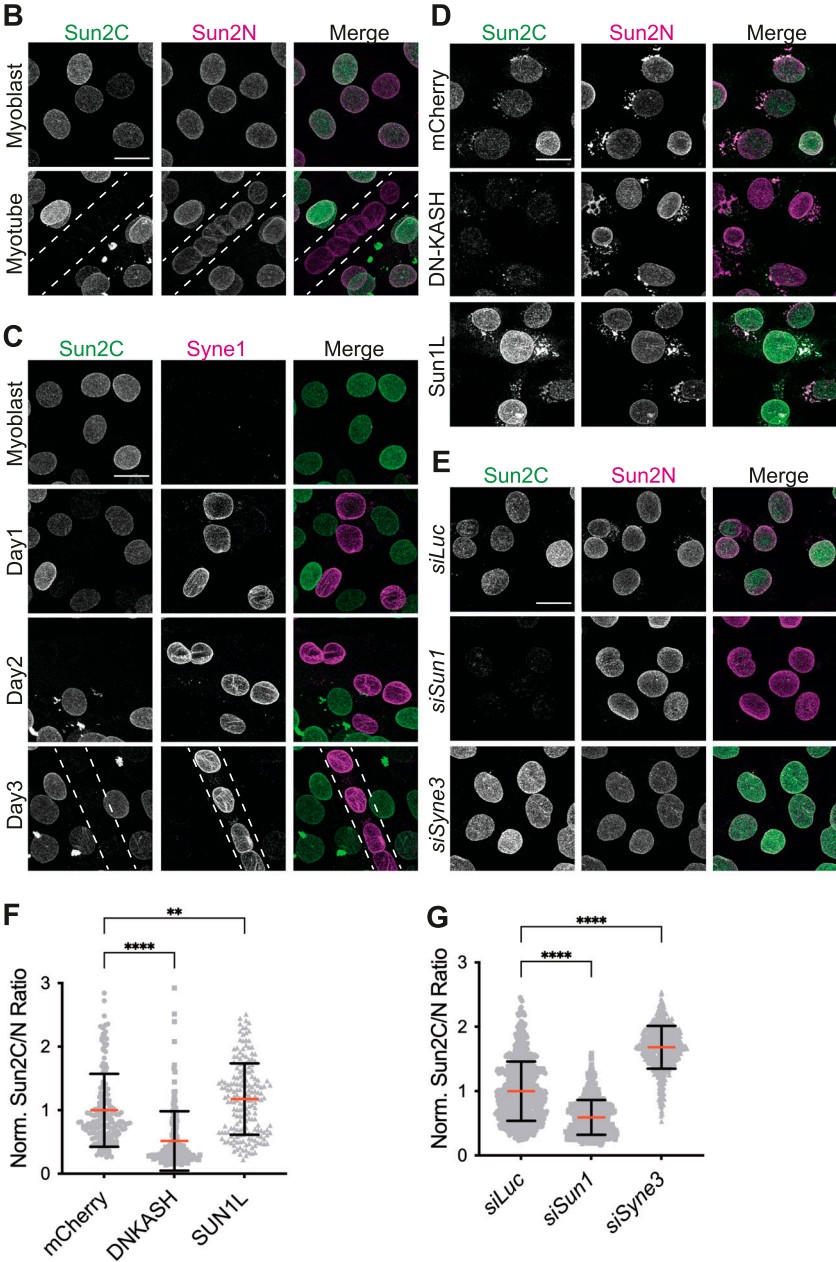

**Figure 1.  Nesprin interaction masks the epitope of a monoclonal SUN2 antibody.**
**(A)** Schematic representation of the LINC complex at the nuclear envelope. **(B)** Representative images show Sun2C and Sun2N antibody staining in C2C12 myoblast cells (MB) or myotubes (MT). The dotted line represents MT outline. **(C)** Representative images of C2C12 MB at different stages of differentiation (Day 1–3) into MTs co-stained for SUN2 and NESPRIN1/SYNE1. **(D)** Representative images of C2C12 MB cells stained with Sun2C and Sun2N Ab, 24 h posttransfection with mCherry, DN-KASH, and SUN1L constructs. **(E)** Representative images of C2C12 MB cells 72 h posttransfection with siRNAs against *Luciferase*, *Sun1* or *Nesprin3/Syne3*, co-stained with Sun2C and Sun2N antibodies. **(F, G)** Quantitation for images in (D, E), respectively. **(F, G)** Scatter plot shows the ratio of total nuclear fluorescence intensity of Sun2C over Sun2N for individual cells in different conditions and normalized to the average ratio of mCherry (for (F)) or siLuc (for (G)). Mean (Red bar) and SD (black bar) are represented. Data pooled from two independent experiments. n > 100 (for (E)) and n > 600 (for (F)) cells per condition. **(F, G)** $t$ test was applied to calculate statistical significance against mCherry (for (F)) and siLuc (for (G)). *$P < 0.05$, **$P < 0.01$, ****$P < 0.0001$. Values show the mean percentage of that group. All images are max intensity projections of confocal z-stacks. All scalebars are 20 $\mu m$.

emerging model describes a heterohexameric complex wherein SUN homotrimer interacts with three KASH peptides (Sosa et al, 2012; Wang et al, 2012). SUN oligomerization has been shown as a prerequisite for KASH interaction but the principles behind SUN oligomerization are still being actively investigated (Jahed et al, 2021). One study suggests that an interplay between an activator CC1 and inhibitor CC2 domain regulates SUN2 monomer to trimer ratios and thereby, SUN–KASH interactions (Nie et al, 2016). Another recent study proposes a self-locking state wherein two SUN trimers are locked head-to-head and unlock at the INM for KASH interaction (Cruz et al, 2020). Conversely, an experimentally validated intermolecular disulfide bridge between cysteine residues in SUN and KASH domains has been described that is dispensable for KASH binding but required for maximal force transmission (Jahed et al, 2015; Cain et al, 2018). Lastly, two conserved C-terminal cysteines in SUN2 have been predicted to form an intramolecular disulfide bond (Sosa et al, 2012). However, the importance of these cysteines remains to be evaluated. An obvious limitation of these structural studies is the use of truncated fragments of proteins in non-physiological conditions. Thus, several key questions remain unanswered: do these autoinhibitory conformations exist in vivo? How do SUN proteins remain inactive and KASH unbound in ER but activate after reaching INM? What controls the SUN activation step? How do LINC complexes respond to dynamic changes in mechanical forces? Lack of appropriate tools remains a major limitation to investigate these questions.

Here, we have identified a key structural change in SUN2 associated with the assembly and disassembly of LINC complex in situ. We report the discovery of a conformation-specific SUN2 antibody that specifically recognizes a SUN2 fraction that is not bound to KASH. Using a semiquantitative imaging assay, we show that KASH bound and unbound SUN2 fraction changes during cell proliferation, differentiation, and migration. Molecular dissection of the antibody epitope revealed a key structural feature of the SUN–KASH interaction, namely that KASH interaction oxidizes the highly conserved SUN2 cysteine residues, thereby masking the epitope. Moreover, genetic and pharmacological perturbation experiments provide evidence for the functional conservation of SUN2 cysteine residues and identify potential redox regulators present in the ER. Overall, we provide a structure–function analysis for SUN2 proteins in a physiological context.

## Results

### Discovery of a conformation-specific SUN2 antibody

Muscle differentiation changes the overall NE proteome and additionally alters the stoichiometry of LINC complex proteins (Chen et al, 2006; Randles et al, 2010; Wilkie et al, 2011; Loo et al, 2019). To understand changes in these proteins during myogenesis, we performed immunofluorescence (IF) staining in C2C12 myoblasts (MB) and terminally differentiated myotubes (MT) using two commercially available highly specific monoclonal antibodies against the C- terminus (Sun2C, #ab124916; Abcam) and N-terminus (Sun2N, #MABT880; Emd Millipore) of SUN2, which we validated by

siRNA knockdowns (Fig S1A–C). Interestingly, the two antibodies behaved differently. The antibody Sun2N, targeting the N-terminus epitope of SUN2, resulted in homogenous nuclear rim staining confirming the presence of SUN2 in both MBs and MTs (Fig 1B, Sun2N). In contrast, we observed a striking heterogeneity of nuclear staining in MBs and failed to obtain a Sun2C signal in MTs altogether (Fig 1B, Sun2C). These results suggest that the epitope of Sun2C is masked in MT.

Onset of myogenesis is marked by the up-regulation of an MT-specific NESPRIN1/SYNE1 that is required for muscle differentiation and fundamentally alters LINC complex composition in MT (Espigat-Georger et al, 2016; Stroud et al, 2017). Therefore, we tested whether the gain of SYNE1 is associated with the loss of Sun2C epitope in MTs. Upon co-staining MBs at different stages of differentiation, we found that the loss of Sun2C epitope coincided with SYNE1 up-regulation as early as Day 1 and remained inaccessible thereafter (Fig 1C). Because SYNE1 is a binding partner of SUN proteins, this suggested a link between NESPRIN up-regulation and epitope masking of the Sun2C antibody in MTs.

Differentiation changes LINC complex composition, but whether the masking of Sun2C epitope in MTs is dependent on differentiation or represents an independent change in LINC complex remained unclear. To address this question, we mimicked NESPRIN up-regulation in undifferentiated MBs by transiently expressing a dominant negative form of NESPRIN (DN-KASH) that lacks the actin-binding domain but retains the KASH domain (Stewart-Hutchinson et al, 2008; Lombardi et al, 2011). Upon co-staining with both SUN2 antibodies, we found that Sun2C failed to recognize SUN2 in the presence of DN-KASH compared with mCherry alone expressing cells (Fig 1D) with significantly reduced Sun2C intensity (Fig 1F, see the Materials and Methods section for quantitation). This shows that Sun2C epitope masking is independent of differentiation but dependent on SUN2:DN–KASH interaction. Structural data show that SUN:KASH interaction can be isoform specific (Cruz et al, 2020). Therefore, we tested a DN-KASH mutant that resembles KASH3 and found that both KASH1 and KASH3-like peptides resulted in Sun2C epitope masking (Fig S2A and B). We hypothesized that, if the Sun2C epitope is masked in KASH-bound SUN2, then conversely, the Sun2C epitope should be unmasked in KASH-unbound SUN2. To test this, we transiently expressed SUN1L, a previously reported luminal version of SUN1 that outcompetes endogenous SUN proteins for KASH binding (Stewart-Hutchinson et al, 2008). As expected, cells expressing SUN1L showed enhanced binding of Sun2C and significantly increased intensities (Fig 1D and F). Overall, this experiment shows that Sun2C epitope is masked in a SUN2:KASH complex but remains accessible for unbound SUN2.

SUN1 and SUN2 both compete for limited KASH peptides at the NE. We hypothesized that in the absence of SUN1, more NESPRIN molecules are available for SUN2 binding, thereby masking the Sun2C epitope. Similarly, loss of NESPRIN would result in SUN proteins being unbound rendering the Sun2C epitope to be accessible. To confirm that Sun2C Ab responds to changes in SUN2–KASH interactions, we transiently depleted either SUN1 or NESPRIN3/SYNE3 in MBs (Fig S1D–F) and co-stained with SUN2 antibodies. As expected, depleting SUN1 decreased and depletion of NESPRIN3 increased Sun2C intensity (Fig 1E and G). Taken together, these data indicate that Sun2C is a conformation-specific SUN2 antibody that only

recognizes unbound SUN2 molecules, and by using this antibody, we show that SUN2 exists predominantly in a KASH-bound state in MTs.

### KASH interaction alters intermolecular disulfide bonds in SUN2

KASH-dependent Sun2C epitope masking presents a rare and interesting opportunity to investigate the LINC complex assembly and dynamics in a physiological context. In vitro structural studies have revealed SUN2 oligomerization as a prerequisite for KASH engagement (Sosa et al, 2012). Therefore, we wondered whether the progressive loss of Sun2C signal observed during myogenesis correlates with changes in SUN2 oligomerization. To test this, we biochemically resolved disulfide-linked SUN2 oligomers from cells undergoing differentiation, using a previously published nonreducing PAGE method (Lu et al, 2008). We found that undifferentiated MBs contained SUN2 monomers (Sun2[M]) and higher molecular weight oligomers (Sun2[O]). In contrast, fully differentiated Day 6 MTs lost most of the SUN2 oligomers (Fig 2A –DTT, Fig 2B). In addition, all oligomers were lost upon treatment with a reducing agent, DTT (Fig 2A +DTT), confirming that the higher molecular weight bands are linked by a disulfide bond. We conclude that SUN2 undergoes an intermolecular disulfide bond rearrangement during C2C12 differentiation.

Previously, it has been shown that SUN2 forms a disulfide bond with NESPRINs to increase the structural integrity of LINC complexes (Sosa et al, 2012; Cain et al, 2018). To test whether KASH interaction leads to loss of disulfide-linked SUN2 oligomers, we transfected MBs with previously described DN-KASH, SUN1L, and mCherry control constructs (Lombardi et al, 2011). Interestingly, in our biochemical analysis, we found that SUN2 oligomers (Sun2(O)) significantly decreased in the presence of DN-KASH and an additional band representing disulfide-linked SUN2:DN-KASH was detected (Fig 2C –DTT and Figs 2D and S2C). The SUN2 oligomer band also decreased upon SUN1L expression but remain unaffected in mCherry controls (Fig 2C –DTT and Figs 2D and S2C). Again, all oligomers were lost under reducing conditions (Fig 2C +DTT). These data suggest that the assembly and disassembly of the SUN2–LINC complex is associated with changes in SUN2 disulfide linkages.

To further address the possibility that the higher molecular weight band represents SUN2 homo-oligomer or the SUN2:KASH complex, we compared the assembly of GFP-tagged full-length SUN2 alongside a SUN2 deletion mutant that disrupts NESPRIN binding (Fig S3A and B). We stably expressed these constructs in C2C12 MBs and performed biochemical analyses to compare disulfide linkages of different SUN2 mutants. We found that N-terminal GFP-tagged full-length SUN2 behaved similar to endogenous SUN2 and showed a single higher molecular weight band (Fig S3B, bands: a/a*, c/c*). In contrast, C-terminal GFP-tagged SUN2 and a SUN domain deletion mutant showed multiple higher molecular weight bands on the blot (Fig S3B, band: b/b*). This suggests that SUN2 mutants that lack nesprin binding are still capable of generating misassembled homo-oligomers and supports the idea that the disulfide-linked higher molecular weight bands on nonreducing gels are SUN2 homo-oligomers rather than the SUN2–NESPRIN complex. Lastly, to test whether

these disulfide rearrangements observed on the gel are physiologically relevant, we treated MBs and MTs with DTT before fixation and performed IF with Sun2C and Sun2N Abs, which was sufficient to retrieve the Sun2C epitope in MTs and significantly increased Sun2C intensity compared with control (Fig 2E and F). This further confirms that the SUN2–KASH interaction leads to formation of disulfide bridges that mask the epitope for Sun2C in situ.

### Conserved C-terminal cysteines determine SUN2 NE targeting, protein turnover, and Sun2C epitope masking

Because disulfide bonds are formed between cysteine residues, we focused on the AA composition of SUN2 to unravel the principles underlying SUN2:KASH-mediated disulfide linkages and Sun2C epitope masking. Mouse SUN2, which is a 731 AA protein, contains only three highly conserved cysteine residues that reside in the SUN domain (Fig S4A and B). Structural data show that SUN2 Cys577 (Cys563 in humans) interacts with Cys-23 in the KASH domain of NESPRINs, whereas Cys615 (Cys601 in humans) and Cys719 (Cys705 in humans) form an intramolecular disulfide bridge (Fig 3A) (Sosa et al, 2012; Wang et al, 2012; Cruz et al, 2020). To understand which cysteines are involved in Sun2C masking, we generated N-term GFP-tagged SUN2 (WT) and alanine-substituted cysteine mutant (C577A and C719A) constructs. Upon stable expression in MBs, we observed that WT- and C577A-expressing lines showed high GFP expression (Fig 3B GFP panel, Fig 3C and D). The ectopically expressed protein localized properly to the NE similar to endogenous SUN2 (Fig 3B and E). On the contrary, the C719A mutant showed comparatively lower expression and accumulated predominantly in the ER (Fig 3B–E).

To identify the cysteines responsible for Sun2C epitope masking, we performed IF with Sun2C antibody in these transgenic lines and found that C719A showed greater binding of Sun2C Ab compared with WT or C577A with significantly higher Sun2C intensity (Fig 3B and F). Lastly, to precisely map the Sun2C epitope, we expressed a C-terminal 17 AA deletion mutant (1–714 AA) and found that Sun2C failed to recognize this construct (Fig 3B). Taken together, these data show that Sun2C Ab recognizes the last 17 AAs of SUN2 that harbors the C719 residue (Fig S4A and B). Enhanced binding of Sun2C Ab to C719A indicates that in in situ experiments, Sun2C exclusively binds to SUN2 molecules with reduced cysteines at 719 and 615.

Disulfide bonds in proteins help with the folding and stabilization of specific conformations (Mamathambika & Bardwell, 2008). To address the impact of cysteine mutation on SUN2 protein stability, we compared the protein turnover of SUN2 WT and mutants by treating cells with cycloheximide (CHX), which inhibits protein synthesis to determine the residual protein at different time points by immunoblotting. Endogenous and overexpressed SUN2 WT degraded at similar rates and about 50% protein remained after 8 h (Fig 3G and H black and red lines). Comparatively, both cysteine mutants, C719A and C577A degraded much faster with only 20% and 30% protein remaining after 8 h, respectively (Fig 3G and H, green and blue lines), suggesting that although both cysteines contribute to maintaining SUN2 stability, C615–C719 disulfide bridge is critical

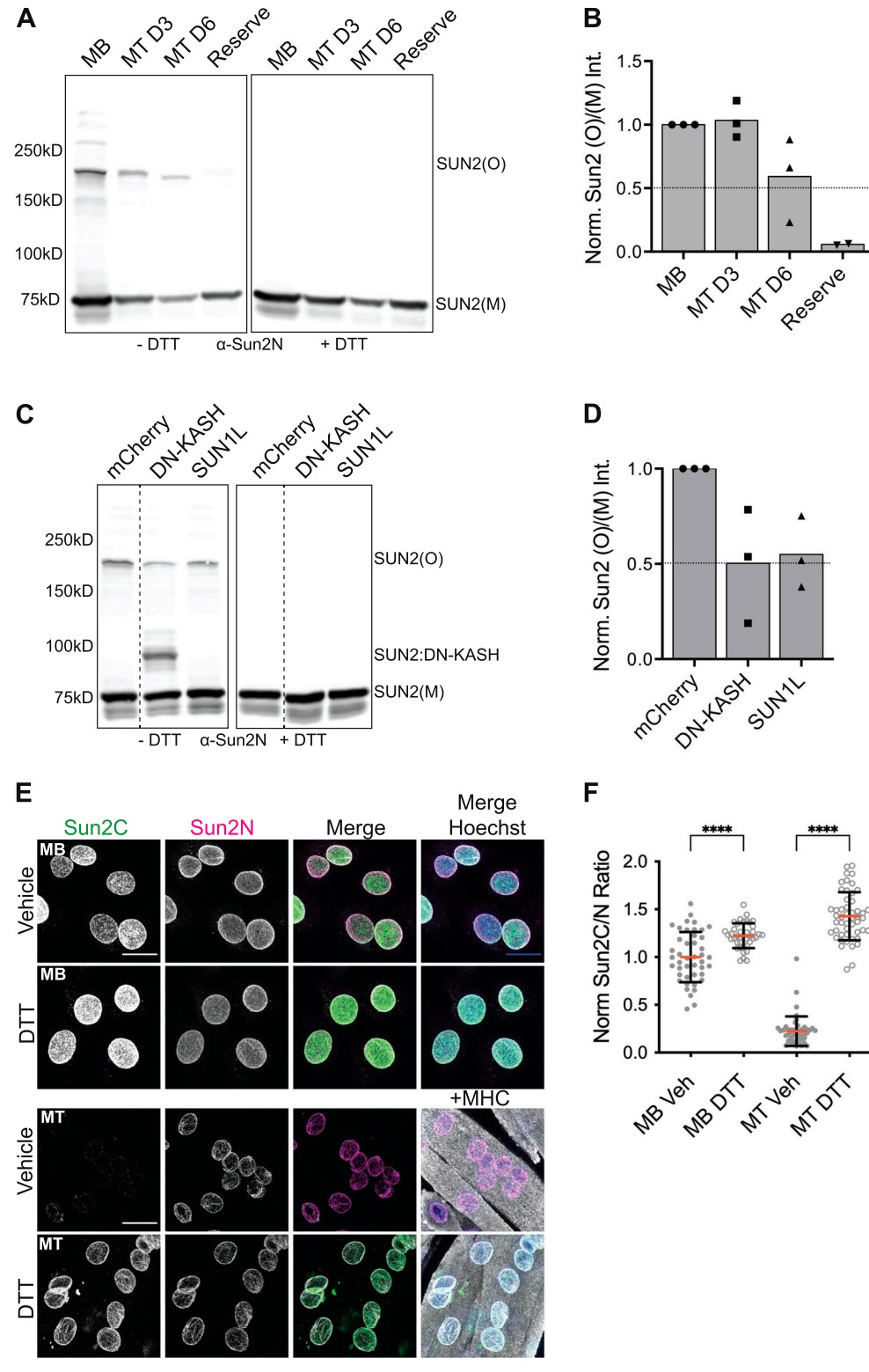

**Figure 2. Nesprin interaction changes SUN2 disulfide linkage.**
**(A)** Western blot shows differential migration of SUN2 monomer (SUN2(M)) and oligomer (SUN2(O)) bands in C2C12 myoblast (MB), Day 3, and Day 6 myotubes (MT D3, D6) and Reserve cells in nonreduced (left panel, –DTT) or reduced (right panel, +DTT) conditions. **(C)** Similar to (A), lysates from MBs, 24 h posttransfection of mCherry, DN-KASH or SUN1L constructs. A separate band for SUN2:DN–KASH complex is observed at ~100 kD. Blot was spliced vertically along the dotted line to remove an unwanted lane. **(B, D)** show densitometry analysis of (A, C) respectively. **(B, D)** Bar plot represents the ratio of SUN2(O) to SUN2(M) band intensity per condition; normalized to MB (for (B)) and mCherry (for (D)). Replicates are shown in different symbols. **(E)** Representative images of MB or MTs treated with Vector (Water) or 5 $\mu$M DTT and co-stained with Sun2C and Sun2N antibodies. Myosin heavy chain staining was used as a MT marker. **(F)** Quantification for (E) scatter plot shows the ratio of total nuclear fluorescence intensity of Sun2C over Sun2N signal for individual cells in different conditions and normalized to the average ratio of Vector. n > 40 cells per condition. Mean (Red bar) and SD (black bar) are represented. $t$ test was applied to calculate statistical significance against corresponding Vehicle control. *$P < 0.05$, **$P < 0.01$, ****$P < 0.0001$. All images are max intensity projections of confocal z-stacks. Scalebar is 20 $\mu$m.

for SUN2 protein turnover. Taken together, our data show that the SUN2 terminal disulfide bridge (C615–C719) shields Sun2C Ab epitope that can be revealed upon disrupting this bond (Fig 3I). More importantly, we show that this bond is required for proper localization and turnover of SUN2, thereby providing evidence for functional conservation of Cys615 and Cys719 residues.

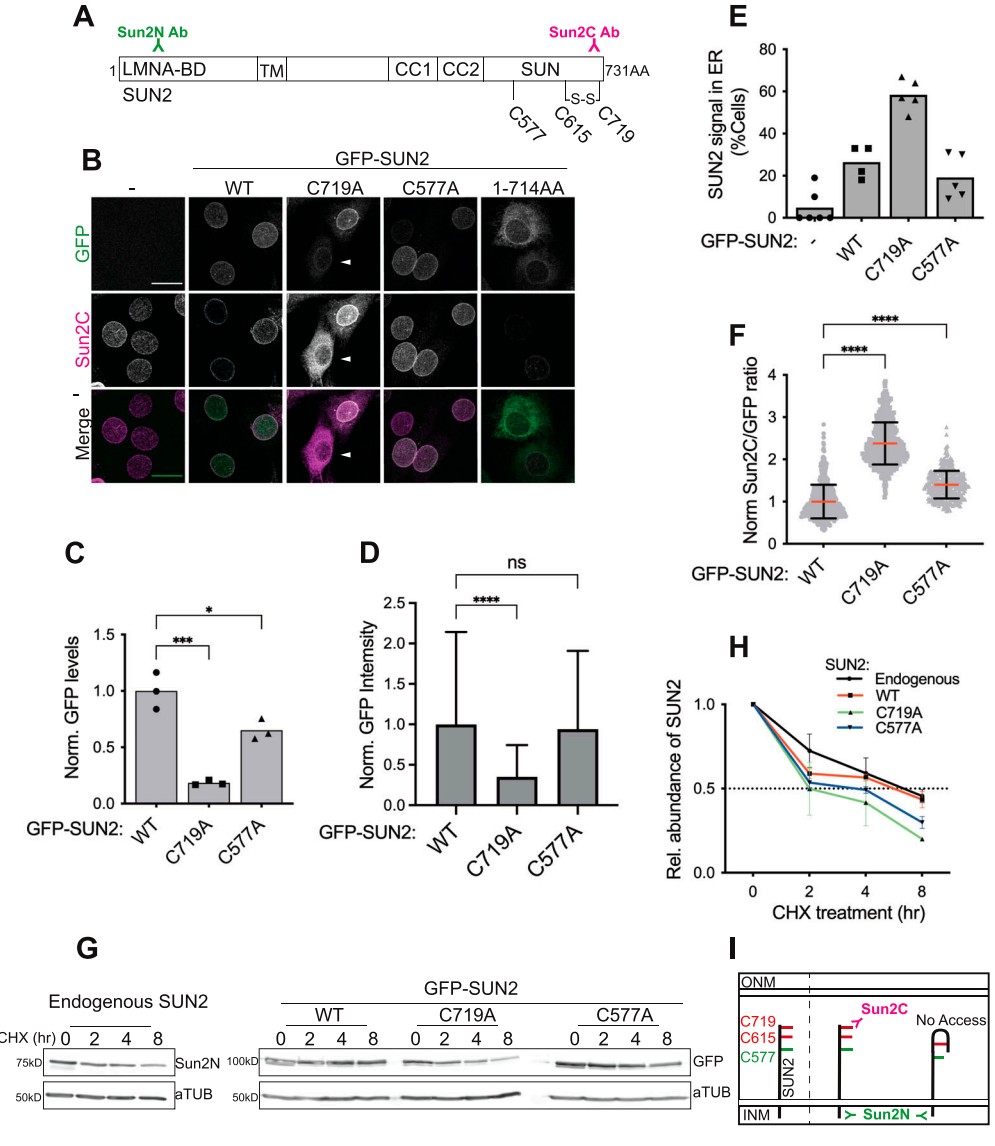

Figure 3. SUN2 terminal disulfide bond is required for proper NE localization and turnover of SUN2.
(A) Schematic diagram depicting different domains within mouse SUN2 full-length protein (1–731 AA): Lamin A-binding region (LMNA-BD), transmembrane domain (TM), coiled coil1 and 2 (CC1 and CC2), SUN domain (SUN). Three cysteines present in SUN2 are labelled according to their amino acid positions in the peptide (C577, C615, C719). S–S represents a disulfide bond. Sun2N and Sun2C represents the corresponding epitopes in the SUN2 peptide for their respective antibodies (also see Fig S4A and B). (B) Representative images of C2C12 cells stably expressing N-terminal GFP-tagged SUN2 (WT) or alanine-substituted cysteine mutants (C719A and C577A) stained with Sun2C antibody. In addition, a mutant with 17 AA deletion at SUN2 C-terminus (1–714 AA) is included that does not recognize Sun2C Ab. The arrowhead indicates the cells expressing low levels of the C719A mutant. (C) Bar graph shows the total GFP level (normalized to a-Tubulin) in GFP-SUN2 WT and mutant lines based on densitometry analysis of Western blots. Each symbol represents an individual replicate of three independent experiments. (D) Bar graph shows mean value with SD of GFP fluorescence intensity in images acquired for GFP-SUN2 WT and mutant lines. n > 400 cells per condition across two independent repeats. (E) Bar graph shows the percentage of cells with SUN2 signal in ER for different cell lines. Each symbol represents a different field of view from two independent days of imaging. n > 75 nuclei per condition. (F) Refers to images in (B). Scatter plot shows the ratio of total nuclear fluorescence intensity of Sun2C over GFP for individual cells from different mutant lines and normalized to the average ratio of WT. Data pooled from two independent experiments. n > 400 cells per condition. t test was performed to measure significance against WT.

(G) Western blot shows the level of SUN2 (endogenous or mutants) and a-Tubulin in a cycloheximide chase experiment. (H) Line plot shows relative abundance of SUN2 (normalized to a-Tubulin) in a cycloheximide time course experiment. Data obtained from the densitometry analysis of protein bands on Western blots probed with Sun2N (for endogenous SUN2) or GFP (for GFP-SUN2 WT and mutants) antibodies. Each cell line is represented by a differently colored line and data normalized to the mean value at 0 h for that cell line. Symbols represent mean values of three independent experiments and bars represent SD. (I) Schematic diagram shows that the Sun2C epitope is inaccessible in the presence of a disulfide bond between Cys615 and Cys719 and only becomes accessible when the bond is broken. For all scatter plots, red bar shows mean and black bar shows SD. Statistical significance represented as: *P < 0.05, **P < 0.01, ****P < 0.0001. Values show mean percentage of that group. All images are max intensity projections of confocal z-stacks. All scalebars are 20 µm.

## SUN2 cysteines contribute to proper assembly of the LINC complex

Because SUN proteins are an integral part of the LINC complex, we asked whether compromising SUN2 stability upon cysteine mutation had an impact on the associated LINC complex (Ketema et al, 2007; Stewart-Hutchinson et al, 2008; Lombardi et al, 2011). To test this, we transiently depleted endogenous SUN2 and stained for other LINC complex proteins, in WT- and mutant-expressing lines. In C2C12 MBs not expressing any transgene, SUN2 depletion had no effect on SYNE3, EMERIN, and SUN1 (Fig 4A and B), suggesting that

SUN1 likely compensates for the loss of SUN2 to maintain LINC complexes at the NE. Similarly, overexpression of WT or C577A showed negligible effect. However, we observed a significant loss of SYNE3, EMERIN, and SUN1 at the NE in cells overexpressing the C719A mutant (Fig 4A and B), indicating that C719A acts as a dominant negative to disrupt LINC complexes.

Next, we addressed which of the SUN2 cysteines participate in disulfide-linked SUN2 oligomerization and KASH binding. First, we resolved disulfide-linked SUN2 homo-oligomers from SUN2 WT and cysteine mutants and found that the C577A mutant lead to a complete loss of SUN2 higher molecular weight band (Fig 4C,

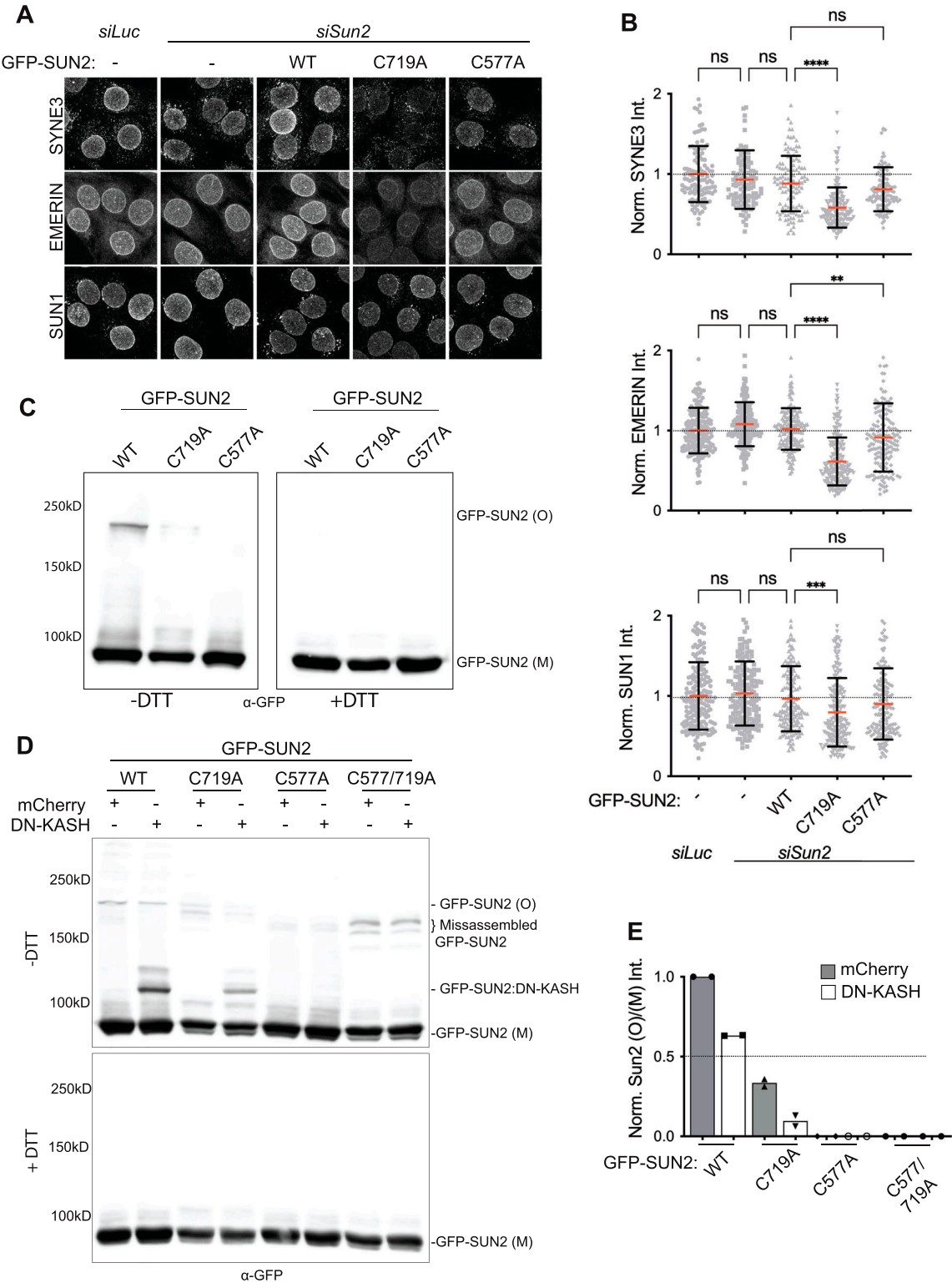

**Figure 4. SUN2 cysteines regulate proper assembly of the LINC complex.**
**(A)** Representative confocal images of C2C12 cell lines stably expressing SUN2 WT or mutant constructs, treated with *siLuc* or *siSun2* and stained for SYNE3, EMERIN or SUN1. **(B)** Quantitation for images in (A). Scatter plot represents total nuclear fluorescence intensity for individual cells across different cell lines. Data pooled from two independent experiments, n > 100 cells per condition. One-way ANOVA with multiple comparisons. For scatter plots, red bar shows mean and black bar shows SD. Statistical significance represented as: *$P < 0.05$, **$P < 0.01$, ****$P < 0.0001$. Values show the mean percentage of that group. All images are max intensity projections of confocal z-stacks. Scalebar is 20 $\mu$m. **(C)** Western blot shows differential migration of GFP-tagged SUN2 cysteine mutants on nonreducing gels. **(D)** Western blot shows

SUN2(O)) compared with WT or C719A mutant suggesting that C577 is critical for the assembly of disulfide-linked SUN2 homo-oligomers. Conversely, the C719A mutant showed multiple faint bands suggesting that although C577 is intact, the terminal disulfide bond between C615–C719 participates in proper assembly of C577-linked SUN2 homo-oligomers.

Second, we overexpressed mCherry and DN-KASH constructs in GFP-SUN2 WT and cysteine mutant lines and found that WT showed a single oligomer band and the band intensity decreased upon DN-KASH expression similar to endogenous SUN2 (Fig 4D and E). Interestingly, the SUN2 C719A mutant was able to interact with DN-KASH peptide (Fig 4D) and showed a decrease in the disulfide-linked homo-oligomer intensity (Fig 4D and E). Because C719A mutant fails to reach the nuclear envelope (Fig 3E), we consider these interactions nonphysiological. On the contrary, the C577A mutant completely abolished KASH interaction (Fig 4D), as has been previously reported (Cain et al, 2018). In addition, we included a SUN2 cysteine double mutant (C577/719A) that has a single cysteine residue (Cys615). As expected, this mutant failed to interact with DN-KASH peptide but showed bands corresponding to misassembled dimers that ran at a lower molecular weight (Fig 4D and E) compared with the single band of properly assembled SUN2 WT homo-oligomers. These data additionally support our previous observation that the higher molecular weight band is a disulfide-linked SUN2 homo-oligomer. Overall, our data point towards the importance of these conserved cysteines in proper cellular localization and assembly of LINC complexes specifically at the NE.

### SUN2 cysteines regulate the actin cytoskeleton

LINC complexes directly associate with the cytoskeleton to balance mechanical forces across the cell and thereby influence cytoskeletal remodeling (Lombardi et al, 2011; Stewart et al, 2015; Gimpel et al, 2017). To address the role of SUN proteins in regulating actin cytoskeletal organization, we transiently depleted SUN proteins in C2C12 MBs and visualized polymerized actin (F-actin) (Fig 5A). Surprisingly, we observed that any perturbation to either SUN1 or SUN2 caused a significant decrease in F-actin intensities (~32% loss), which declined even further upon loss of both SUN proteins (Fig 5A and B). These data suggest that F-actin is sensitive to the level of SUN proteins and that neither of the SUN proteins can compensate for maintaining proper F-actin levels. Next, to test whether SUN2 cysteines additionally contribute to cellular F-actin regulation, we transiently depleted endogenous SUN2 and performed a rescue experiment with GFP-SUN2 WT and cysteine mutant lines (Fig 5C). We observed a 24% loss of total F-actin levels compared with *siLuc* control; however, the expression of GFP-SUN2 WT did not show a significant rescue (Fig 5D). Interestingly, the expression of both cysteine mutants, C719A and C577A not only failed to rescue but further enhanced the F-actin phenotype by

decreasing the F-actin levels to 45% and 52%, respectively (Fig 5D); comparable with *siSun1/2* levels. Taken together, our data show that actin cytoskeleton regulation is not only sensitive to the total level of SUN proteins but also depends on their redox state.

### Dynamic changes in SUN2 cysteine oxidation state regulate cell migration

The LINC complex plays an important and well-documented role in cell migration during wound healing of a cell monolayer (Gant Luxton et al, 2010; Lombardi et al, 2011; Chang et al, 2015; Zhu et al, 2017; Cain et al, 2018). We took advantage of wound healing/scratch assay to investigate the dynamics of SUN2 terminal disulfide bond during directional cell movement. To test for disulfide bond rearrangements, we performed IF with Sun2C and Sun2N Abs at different time points post wound. We observed a distinct loss of Sun2C staining specifically in migrating cells at the edge of the wound compared with confluent monolayers as early as 6 h post wound (Fig 6A). We quantified these changes and observed a significantly lower Sun2C intensity at the wound edge 8 h post scratch compared with 0 h time point (Fig 6B and C Edge), whereas Sun2C intensity remained unchanged away from the scratch site (Fig 6B and C Center). This experiment shows that migrating cells exclusively present at the wound edge undergo dynamic SUN2 terminal cysteine oxidation suggesting a remodeling of the LINC complex.

To directly address the importance of SUN2 cysteines during cell migration, we performed wound healing assays in our SUN2 WT and cysteine mutant lines upon transient depletion of endogenous SUN2. Our data show that cells migrate slower in the absence of SUN2 but recover upon compensating with WT and to a lesser extent with C577A mutant. However, cells overexpressing C719A exhibited significantly slower migration compared with the control *siLuc* (Fig 6D), suggesting that SUN2 terminal disulfide bridge is required for proper cell migration. Consistent with previous reports, our data show that upon receiving migratory cues, cells reorganize their nucleo-cytoskeletal structure and engage their LINC complexes to facilitate directional cell movement. For the first time, these transition states can be clearly visualized using Sun2C antibody at precise spatiotemporal resolution providing molecular insights into cell migration.

### ER microenvironment regulates SUN2 terminal disulfide bond

Our data directly link the redox state of SUN2 terminal cysteines to several molecular and cellular phenotypes. The C-terminus of SUN2 resides in the ER/NE lumen that harbors proteins crucial for protein folding, quality control, intracellular calcium homeostasis, and redox balance (Görlach et al, 2006). Therefore, we tested whether SUN2 cysteines respond to changes in ER homeostasis. To induce ER stress, we transiently treated C2C12 MBs with either an ER $Ca^{2+}$

---

differential migration of ectopically expressed GFP-SUN2 WT and mutant proteins as monomers (GFP-SUN2 (M)) and oligomers (GFP-SUN2 (O)) in nonreducing (−DTT, top panel) and reducing (+DTT, bottom panel) conditions. GFP-SUN2:DN-KASH complex can be detected as a single band (GFP-SUN2:DN-KASH). **(E)** Quantitation for western images in (D). Bar graph shows the ratio of GFP-SUN2(O) to GFP-SUN2(M) band intensity per condition across SUN2 WT and mutant lines; normalized to WT mCherry expressing cells. Gray bars: mCherry and white bars: DN-KASH expression.

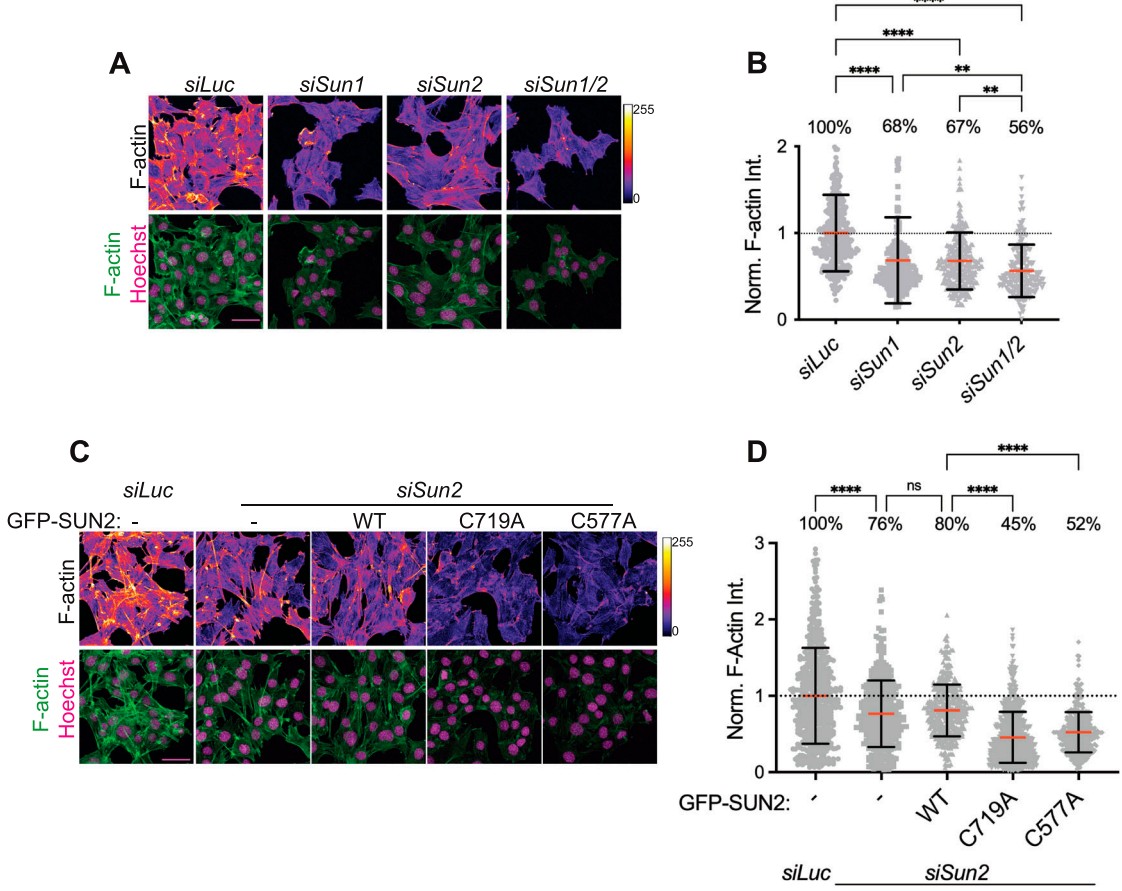

**Figure 5. LINC complexes regulate actin cytoskeletal structure.**
**(A)** Representative confocal images of C2C12 cells, 72 h post treatment with siRNA against *Luciferase* (control), *Sun1, Sun2,* and *Sun1/2*, stained with fluorescently labelled Phalloidin for F-actin (pseudocolored, top panel) and nuclei counterstained with Hoechst (bottom panel). **(B)** Scatter plot represents the distribution of Phalloidin intensities per cell in each condition in (A). **(C)** Representative confocal images of C2C12 cell lines stably expressing GFP-tagged WT or mutant constructs, subjected to *siLuc* or *siSun2* treatment for 72 h and stained with fluorescently labelled Phalloidin for F-actin (pseudocolored, top panel) and GFP (bottom panel). **(D)** Similar to (B). For both (B, D), data pooled from two independent experiments and normalized to average value of *siLuc* condition. n > 230 cells per condition. One-way ANOVA with multiple comparisons was applied to calculate statistical significance. Red bar shows mean and black bar shows SD. Statistical significance represented as: \**P* < 0.05, \*\**P* < 0.01, \*\*\*\**P* < 0.0001; and ns, not significant. All images are max intensity projections of confocal z-stacks. Scale bar is 50 μm. F-actin was pseudocolored with FIRE lookup table and intensity scale is shown on the right side.

ATPase inhibitor, thapsigargin (Tg), that causes loss of calcium from ER lumen or 16F16 (iPDI), a protein disulfide isomerase (PDI) enzyme inhibitor that compromises protein folding in ER (Hoffstrom et al, 2010). Upon IF with Sun2C and Sun2N Abs, we observed a significant increase in the Sun2C signal in Tg (Fig 7A and B) and iPDI (Fig 7C)-treated cells compared with DMSO control. These changes were observed within 1 h of treatment and did not change with increase in treatment time suggesting that SUN2 terminal cysteines are highly sensitive to ER environment and quickly undergo reduction upon perturbing ER homeostasis.

Multiple studies have implicated another ER resident protein TORSIN1A (TOR1A) in remodeling LINC complexes at NE (Nery et al, 2008; Saunders et al, 2017); however, the molecular mechanisms remain unclear. TOR1A is part of the AAA+ ATPase superfamily of protein that specifically resides in ER/NE lumen and acts as a molecular chaperone to assemble protein complexes (Burdette et al, 2010; Laudermilch & Schlieker, 2016). Therefore, we asked

whether TOR1A participates in maintaining the redox state of SUN2 terminal cysteines. We transiently depleted TOR1A in C2C12 MBs (Fig 7D) and upon co-staining with both SUN2 antibodies, we observed a significant increase in Sun2C signal in TOR1A-depleted cells (Fig 7E and F). This suggests that loss of TOR1A leads to reduction of SUN2 terminal cysteine residues. Moreover, we observed a loss of C577-linked SUN2 oligomers in TOR1A-depleted cells (Fig 7G and H). To evaluate the functional significance of these observations, we performed a wound healing assay and found that TOR1A-depleted cells migrated slower and showed a significant decline in wound closure compared with control (Fig 7I). These data are consistent with a previous study (Nery et al, 2008) and supports our own observations showing relatively slower migration of C719A lines lacking SUN2 terminal disulfide bond. Taken together, our data show that the redox state of terminal cysteines in SUN2 could dynamically be altered, not only by TOR1A, but by additional factors that regulate ER homeostasis. These results have major

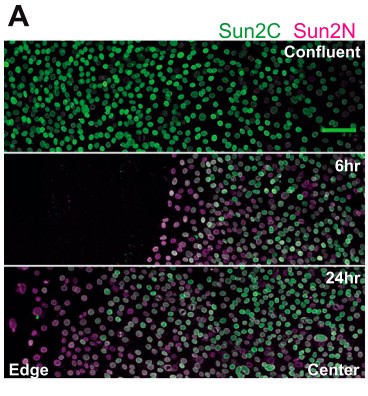

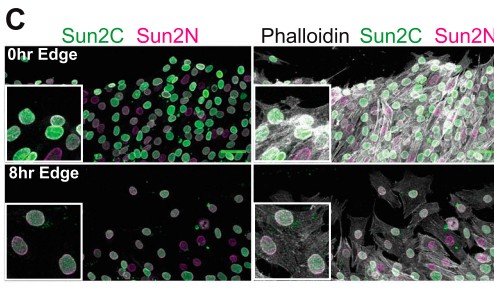

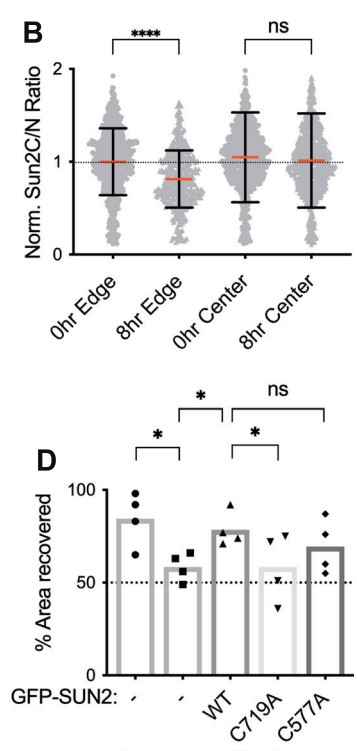

**Figure 6. Dynamic regulation of SUN2 terminal disulfide bond during cell migration.**

**(A)** Stitched confocal image panels of confluent C2C12 cell monolayers (top panel) in a wound healing assay, 6 h (middle panel), and 24 h (bottom panel) post-wound and co-stained with Sun2C and Sun2N Abs. Wounded and unwounded regions are labelled as edge and center, respectively. **(B)** Quantitation for images in (A). Scatterplot shows the ratio of total nuclear fluorescence intensity of Sun2C over Sun2N for individual cells at 0 h or 8 h post wound, at the edge of the wound (Edge) or unwounded region (Center). Data pooled from two independent experiments where n > 500 cells per condition. Red bar shows mean and black bar shows SD. Statistical significance tested using t test. **(C)** Representative confocal image of wound healing assay. Images taken 0 h and 8 h post wound and stained with Sun2C, Sun2N, and Phalloidin. Inset shows a zoomed section of the image. **(D)** Bar graph shows the mean percentage of the area recovered 24 h post wound in a wound healing assay for different C2C12-engineered cell lines. Each symbol represents individual replicates of four independent experiments. One-way ANOVA was performed with multiple comparisons. Statistical significance represented as: $*P < 0.05$, $**P < 0.01$, $****P < 0.0001$, ns, not significant. All images are max intensity projections of confocal z-stacks. All scalebars are 100 $\mu$m.

implications in further understanding the rapid remodeling of LINC complexes at the NE.

## Discussion

### Conformation-specific SUN2 antibody

Conformation-specific antibodies are indispensable tools for in situ biological research as they provide unprecedented insights into the function of proteins in their physiological environment. Structural proteins like integrins and lamins are excellent examples where conformational epitopes have unraveled structure–function relationships (Bazzoni et al, 1995; Dyer et al, 1997; Ihalainen et al, 2015). Here, we characterize a SUN2 Ab (Sun2C) that has been widely used in literature and show, for the first time, that it is conformation specific. At the molecular level, we find that SUN2:KASH interaction leads to C615–C719 disulfide bond formation in SUN2, that masks the Sun2C epitope. Using a quantitative imaging assay, we provide evidence that Sun2C epitope masking occurs during key cellular processes like cell proliferation, differentiation, and migration. Because LINC complexes have been previously implicated in all these processes (Gant Luxton et al, 2010; Chang et al, 2015; Aureille et al, 2019; Déjardin et al, 2020; Carley et al, 2021), our data complement these studies by providing a new marker for SUN2:KASH interaction in situ and has the potential to generate novel insights from new and existing datasets. Additionally, our findings support the idea that instead of changing total protein levels at the NE; LINC

complexes can rapidly contribute to biological function by dynamically altering their conformation.

### SUN2 disulfide bonds and LINC complex formation

In vitro structure data have shown that the SUN2 terminal disulfide bond remains intact in KASH bound and unbound states (Sosa et al, 2012; Wang et al, 2012; Cruz et al, 2020). Our data in MBs reveal that newly synthesized SUN2 in the ER- and KASH-bound SUN2 at the NE do not show any Sun2C staining, providing in vivo confirmation of previous structural data. Interestingly, we observe Sun2C staining at the NE in proliferating MBs that suggests that the terminal disulfide bond is dynamic and alters the redox state at the nuclear periphery. Our functional experiments show that Sun2C staining is enhanced when SUN2 is unbound and devoid of KASH suggesting that the terminal disulfide bond is responsive to the SUN2–KASH interaction and a fraction of SUN2 exists naturally in a KASH-unbound state. The presence of reduced SUN2 cysteines specifically at the NE is intriguing and highlights the importance of C615–C719 disulfide bond. We show that disrupting a single disulfide bond by mutating C719 results in accumulation of the protein in the ER, enhanced rate of degradation, and disruption of LINC complex assembly and function. Moreover, we find improperly assembled disulfide-linked SUN2 C719A homo-oligomers on nonreducing gels. This suggests that terminal disulfide bond is required for proper protein folding in the ER, regulation of protein–protein interactions, and participation in proper INM localization of SUN2. So why are SUN2 cysteines reduced at the NE? Our nesprin-3 KD and SUN1L OE

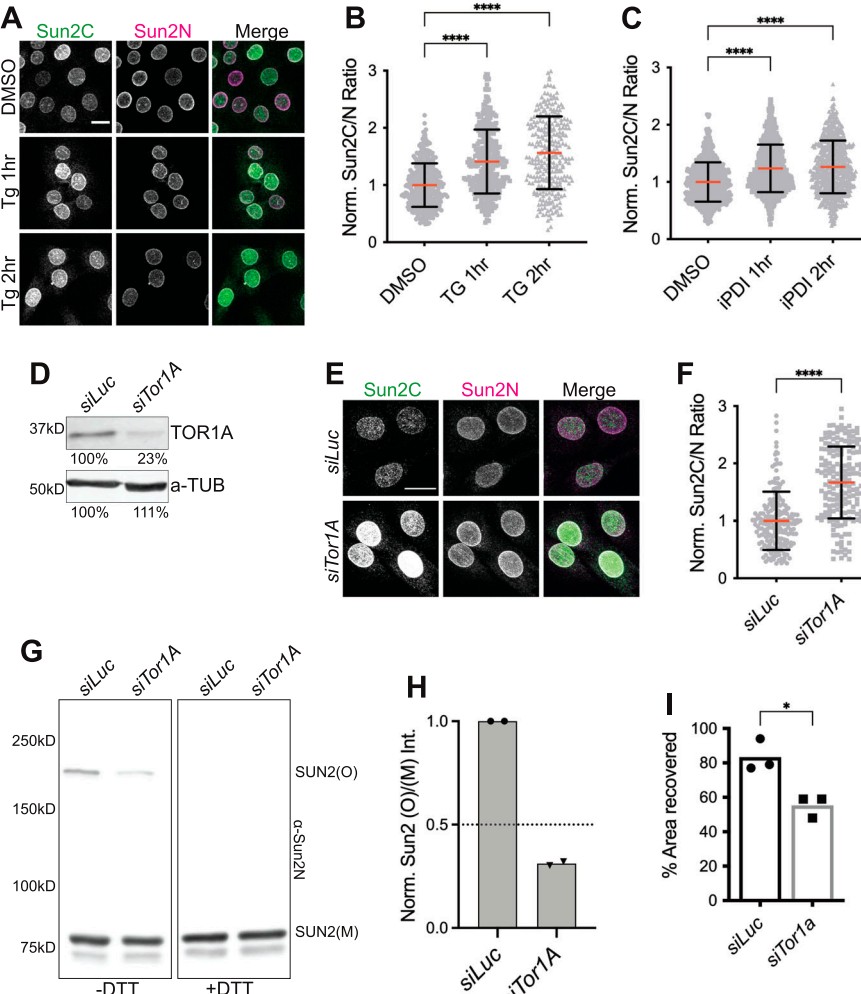

**Figure 7. Endoplasmic reticulum environment regulates oxidative state of the SUN2 terminal disulfide bond.**
**(A)** Representative confocal images of C2C12 MBs treated either with DMSO or thapsigargin (Tg) for 1–2 h and co-stained with Sun2C and Sun2N antibodies. **(B, C)** Scatter plot shows the ratio of total nuclear fluorescence intensity of Sun2C over Sun2N for individual cells in DMSO control, Tg treated (for (B)), and protein disulfide isomerase inhibitor treated (iPDI, for (C)); normalized to the average ratio of DMSO. **(B, C)** Data pooled from two independent experiments. n > 250 (for (B)) and n > 600 (for (C)) cells per condition. **(D)** Western blot against Torsin1A (TOR1A) and a-Tubulin (a-TUB) in luciferase control (*siLuc*) or *Tor1A* (*siTor1A*)-treated MBs. Values show relative band intensity. **(E)** Representative images of *siLuc* or *siTor1A*-treated MBs co-stained with Sun2C and Sun2N antibodies. **(F)** Quantitation for images in (E). Scatter plot shows Sun2C/N ratio in *siLuc*- and *siTor1A*-treated cells normalized to the average ratio of *siLuc*. Data pooled from two independent experiments. n > 180 cells counted per condition. **(G)** Western blot shows SUN2 oligomers (SUN2(O)) and monomers (SUN2(M)) in *siLuc*- and *siTor1A*-treated cells in nonreducing (−DTT) and reducing conditions(+DTT). **(H)** Quantitation for (G). Bar graph shows mean normalized band intensity for SUN2 oligomers in control and knockdown conditions. Replicates are shown in different symbols. **(I)** Bar graph shows the mean percentage of the area recovered 24 h post wound in a wound healing assay for C2C12 cells treated with either *siLuc* or *siTor1A*. Each symbol represents individual replicates of three independent experiments. For all scatter plots, red bar shows mean and black bar shows SD. **(B, C, F, I)** *t* test was applied to calculate statistical significance against controls (DMSO for (B, C); *siLuc* for (F, I)). *P < 0.05, **P < 0.01, ****P < 0.0001. All images are max intensity projections of confocal z-stacks. All scalebars are 20 μm.

experiments show that the lack of KASH engagement is sufficient to reduce the terminal disulfide bond in SUN2, suggesting the presence of an active disulfide rearrangement mechanism at the NE.

At the mechanistic level, it has been shown that SUN2 C577 forms a disulfide bond with KASH C-23 to strengthen SUN–KASH interactions (Sosa et al, 2012; Jahed et al, 2015; Cain et al, 2018). We show the existence of disulfide-linked SUN2 homo-oligomers that disappear upon KASH interaction on nonreducing gels. Using a mutagenesis approach, we identify C577 as the critical residue for disulfide-linked SUN2 homo-oligomerization. Because the same residue is involved in homo-oligomerization and KASH interaction, it is imperative that SUN2 oligomers undergo disulfide bond rearrangement to switch partners. However, structural studies show C577-independent SUN2 oligomerization and KASH engagement (Cain et al, 2018). Therefore, the exact nature of disulfide-linked SUN2 homo-oligomers remains to be further characterized. Considering that only a small fraction of total SUN2 exists as a disulfide-linked oligomer, we think these might be either very short-lived intermediates or remnants of higher-order

branched LINC assemblies connected through C577 disulfide bonds (Gurusaran & Davies, 2021). In addition, coiled coil domains in SUN2 contribute to oligomerization in vitro (Nie et al, 2016), which cannot be resolved by non-reducing gels, and require future investigation. Consistent with previous studies, we find that mutating C577 does not affect protein localization but increases protein degradation that results in a significant effect on LINC complex assembly and function. Most importantly, the C577A mutant shows an increase in Sun2C signal, suggesting that loss of interaction between SUN2 and KASH cysteines does affect the SUN2 terminal disulfide bond. Similarly, the C719A mutant shows misassembled C577-linked homo-oligomers. This shows a dynamic interplay among all three highly conserved cysteines in SUN2 and supports the idea that SUN2 undergoes a disulfide bond rearrangement at the NE during LINC complex formation. Based on these observations, we propose a revised model of LINC assembly and disassembly from the point of view of disulfide bond rearrangement (Fig 8). Overall, our analysis reconciles in vitro data with in vivo observations and strives to fill a major gap in our understanding of LINC complex structure and function in vivo.

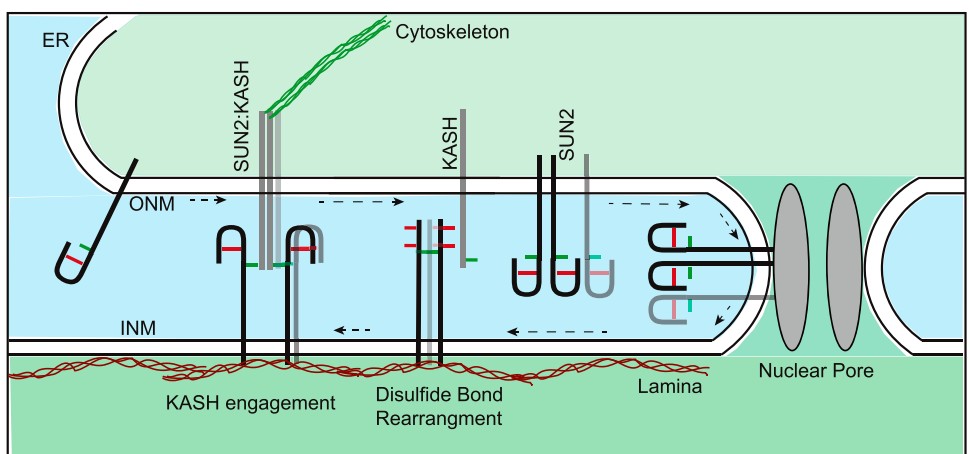

**Figure 8. A proposed model for the LINC complex assembly at the NE.**
Model shows the redox state of SUN2 (black/gray bar) cysteines C719-C615 (red) compared with C577 (green) during transition from ER/ONM to INM (black arrows) and subsequent assembly of SUN2:KASH LINC complex at the INM.

## SUN2 disulfide bonds in cytoskeleton organization and cell migration

LINC complexes interact with the cytoskeleton and affect its organization. Recent literature implicates SUN1 and SUN2 in actin cytoskeletal regulation (Gant Luxton et al, 2010; Stewart et al, 2019; Porter et al, 2020). However, whether this regulation is directly mediated through LINC complex or indirectly through signaling pathways remains unclear. Our data show that loss of either of the SUN proteins decreases polymerized actin levels suggesting that SUN1 and SUN2 play nonredundant roles in actin polymerization in C2C12 MBs and additional pathways might be involved that complicate the phenotype. This might be one of the reasons we observe only a slight but insignificant rescue of F-actin levels upon GFP-SUN2 WT overexpression. Interestingly, we find that both cysteines (C719 and C577) contribute to actin cytoskeleton maintenance as mutating these cysteines further depleted F-actin levels. For C719A, this phenotype can be explained by a concurrent loss in LINC member (SUN1/SYNE3/EMD). However, F-actin loss in C577A was surprising because we did not see much change in nesprins. Interestingly, according to recent reports, SUN2 seem to regulate the RhoA signaling pathway that controls actin polymerization and mechanosignaling, but the mechanisms remain unclear (Thakar et al, 2017; Mu et al, 2020; Porter et al, 2020). This raises a possibility of SUN2 regulating actin independent of LINC complex. Another possibility is that C577A does not affect total levels of nesprin but decreases the residence time of nesprins on the NE, thereby compromising the stability of nesprin–actin interaction (Saunders et al, 2017). Indeed, our data show that C577A degrades faster compared with WT. Taken together, our study points towards the importance of SUN2 cysteine residues and their redox states in the maintenance of proper actin cytoskeleton system, although further experiments are required to understand the mechanisms. Because F-actin is directly related to mechanical stiffness and fluidity of the cell, these findings have implications in cellular transformation, gain of invasive properties, and diagnostic potential in disease conditions (Kalukula et al, 2022).

Directional migration of cell monolayer has been well documented and requires SUN2-dependent LINC complex and actin cytoskeleton (Gant Luxton et al, 2010; Zhu et al, 2017). Our 2D migration assay shows that the SUN2 terminal disulfide bond is intact in cells migrating at the wound edge and disrupting this bond negatively affects migration but does not completely abolish it. These data are consistent with live FRET measurements of migrating cells that show nesprins under increased tension at the leading edge of the wound (Déjardin et al, 2020). Altogether, our work provides visual molecular insights into in vivo LINC complex dynamics during cell migration. We propose, in the future, Sun2C antibody could be used to investigate SUN–KASH engagement during neuronal and muscle development, cancer cell migration, mechanobiology, and other processes where LINC complexes have already been shown to play an important role.

## SUN2 disulfide regulation

Principles behind proper assembly and disassembly of LINC complexes at the NE remain elusive. We show that SUN2–KASH interaction leads to SUN2 terminal disulfide formation and the loss of this bond accelerates SUN2 protein turnover. Structure and molecular simulation dynamics studies have pointed to the importance of pH and ion concentration for in vitro LINC assembly (Jahed et al, 2018; Gurusaran & Davies, 2021). We show that perturbing ER homeostasis is sufficient to rapidly reduce SUN2 terminal cysteines in situ, thereby indicating that ER microenvironment chemically modifies SUN2. Posttranslational modifications of SUN2 have been known to impact its function (Gilbert et al, 2019); therefore, our data suggest an intriguing possibility of a redox-dependent remodeling of LINC complexes. Several studies have pointed to torsins as regulators of LINC complexes. In yeast, overexpressed human SUN1 and SUN2 co-purify with TOR1A suggesting a direct interaction between the two proteins (Chalfant et al, 2019). Interestingly, TOR1A has been shown to form oligomeric helical tube-like structures (Demircioglu et al, 2019). Although these structures have not been shown to use peptides as substrates yet, there might be a possibility that TOR1A oligomers actively cluster SUN proteins to initiate oligomerization and SUN2 disulfide bond rearrangement. TOR1A could also affect NESPRIN distribution on the ONM (Nery et al, 2008). This hypothesis is

supported by our observation that loss of either NESPRIN3 or TOR1A leads to loss of the SUN2 terminal disulfide bond. Alternatively, TOR1A might contribute to SUN2 cysteine oxidation indirectly by regulating ER stress response (Chen et al, 2010). Although we did not observe any increase in ER stress markers (BIP and PDI) upon Tor1A depletion (data not shown), we cannot completely rule out the contribution of other ER stress pathways. Overall, this study lays the foundation for future identification of redox-based LINC complex regulators. Moreover, this work supports the idea that, in addition to their roles in protein folding and stability, some disulfide bridges may possess a key regulatory property that diversifies their functions (Hogg, 2003).

# Materials and Methods

### Cell culture and cell line generation

Mouse C2C12 myoblast cells were cultured in a growth media (DMEM with 20% FBS and 1% Pen-Strep) and passaged every other day. For differentiation to myotubes, confluent C2C12 myoblasts were cultured in differentiation media (DMEM with 2% horse serum and 1% pen–strep) with media changes every other day. Cell lines were cultured under 37°C and 5% $CO_2$ incubator conditions. C2C12 cells stably expressing GFP-SUN2 WT, C719A, and C577A constructs were generated by retroviral transduction of C2C12 cells. For retroviral packaging and production, 293T cells were transfected with 4 µg of Ampho, 4 µg of plasmid of interest and 32 µl of polyethylenimine in 10 cm plates. Virus-containing media were harvested at 48 h and cell-free supernatant was used to transduce early passage C2C12 cells. Successfully transduced cells were selected by culturing in media containing 10 µg/ml Blasticidin for 4 d. Selected cells were expanded in growth media and multiple vials were frozen for future use.

### Plasmids and siRNA

N-terminal GFP-tagged SUN2 WT expression plasmid was constructed as previously described (Buchwalter et al, 2019) using a mouse SUN2 sequence (Uniport ID Q8BJS4). GFP and SUN2 open reading frames were individually amplified using primers with overlapping regions between GFP and SUN2. Forward primer for GFP and reverse primer for SUN2 included attB sites for Gateway cloning. Next, using an overlap PCR strategy, GFP and Sun2 amplicons were stitched together to generate a single amplicon with attB sites. Using a Gateway cloning kit, the GFP-SUN2 amplicon was first cloned into a PDONR207 vector and later into PQCXIB (Campeau et al, 2009), a retroviral backbone gateway destination vector. GFP-SUN2 WT construct was used as a template for cloning all other SUN2 mutant versions. GFP-SUN2(1–714 AA) was amplified using GFP forward primer and a reverse primer containing a stop codon after AA 714 (5'-ctaGTGGCCCCAGTTGGT-CAGGATCC-3') followed by Gateway cloning into PQCXIB vector. We used Quickchange mutagenesis strategy to generate SUN2 cysteine mutants C719A and C577A. We used an overlapping primer set to amplify GFP-SUN2 WT construct by replacing either TGT(C719) or TGC(C577) to GCC (C719A or C577A), thereby substituting alanine for

cysteines. C-terminal-tagged SUN2-GFP construct was previously published (Buchwalter et al, 2019). All constructs were verified by sequencing. pmCherry and pmCherry–C1:KASH1 (DNKASH) constructs were previously published (Hatch & Hetzer, 2016). pmCherry–C1:KASH3 (DN-KASH3) construct was generated by site-directed mutagenesis as mentioned above for SUN2 constructs. CACCCC was changed to GCTCTC to generate the HP to AL AA change in DN-KASH3 (see Fig S2A). A previously published plasmid, pCDH-CMV-MCS-EF1 copGFP-T2A-puro:SS-HA-Sun1L-KDEL (Lombardi et al, 2011), was used to subclone the SS-HA-SUN1L-KDEL region into a pC1 vector to generate pC1:SS-HA-Sun1L-KDEL-IRES-mCherry (SUN1L) construct (plasmid generated by Emily Hatch). pCDH-CMV-MCS-EF1-SS-GFP-KDEL plasmid was a gift from J Lammerding (The Weill Institute for Cell and Molecular Biology, Cornell University) (Lombardi et al, 2011).

Control siRNA against luciferase (siLuc) was custom ordered from Life technologies (sense: 5'-UAUGCAGUUGCUCUCCAGCDTDT-3'). Individual or pooled siRNAs against other targets were ordered from Horizon Discovery (Dharmacon), Lafayette, CO. with the following catalogue numbers: Sun1 (ON-TARGETplus Mouse Sun1 [77053] siRNA—Individual, J-040715-10); Sun2 (ON-TARGETplus Mouse Sun2 [223697] siRNA—Individual, J-041247-09); Syne3 (ON-TARGETplus Mouse Syne3 [212073] siRNA—SMARTpool, L-052180-01); Tor1a (ON-TARGETplus Mouse Tor1a [30931] siRNA—SMARTpool, L-051579-01).

### Transfections and pharmacological treatments

DNA transfection was performed using Lipofectamine 3000 following the manufacturer's protocol directly on cells plated on the previous day, in ibidi chambers (for Imaging) or six well plates (for Biochemistry). Experiments were performed 24 h post DNA transfection. siRNA (50 nM final conc) was reverse transfected in six well plates using Lipofectamine RNAiMAX at 0 h and repeated at 24 h. The cells were split at 48 h and either seeded in ibidi chambers (for imaging) or passaged into six well plates (for protein) to be analyzed at a 72 h time point.

Cells were seeded a day before in ibidi chambers and treated with either DMSO, 100 nM thapsigargin (#T7458; Life Technologies) or 50 µM iPDI (PDI inhibitor 16F16, # SML0021; Sigma-Aldrich) in growth media for 1 or 2 h before fixation and IF. For DTT rescue experiment, cells in ibidi chambers were treated with media (growth for MBs/differentiation for MTs) containing either DI water (Vehicle) or 5 µM final conc. of DTT (#AC16568; Acros Organics) for 3 min before fixation and IF.

For CHX chase experiment, the cells were seeded in six-well plates the previous day and treated either with 200 µg/ml CHX (# C-7698; Sigma-Aldrich) in fresh growth media for 2, 4 or 8 h; or without CHX (0 h). The cells were harvested by trypsinization and flash-frozen cell pellets were stored in –20°C for future protein analysis.

### Immunofluorescence

Cells were cultured in ibidi µ-Slide eight Well (#80826; Ibidi) chambers with indicated treatments. On the day of the experiment, cells were washed once with PBS, fixed in 4% paraformaldehyde in

PBS for 5 min, followed by two PBS washes for 5 min. At this point, the cells were either stored in fresh PBS at 4°C in a humidified chamber for future staining or processed further in the following manner. The cells were permeabilized and blocked in IF buffer (10 mg/ml BSA, 0.1% Triton X-100, 0.02% SDS in PBS) for 20 min, followed by incubation in IF buffer containing primary antibodies at appropriate dilution for 2 h at RT. The cells were washed three times in IF buffer followed by incubation in an IF buffer containing secondary antibodies (and 647 Phalloidin when appropriate) for 1 h at RT. For DNA counterstain, the cells were incubated in IF buffer containing Hoechst for 5 min, followed by three washes with IF buffer. Finally, the cells were washed once with PBS and stored in PBS for confocal imaging the same day.

The following primary antibodies and their respective dilutions were used in IF experiments: mouse Sun1 clone 12.10F (#MABT892; 1: 250; MilliporeSigma); rabbit Sun2(Sun2C) EPR6557 (#ab124916; 1:500; Abcam); mouse Sun2(Sun2N) clone 3.1E (# MABT880,1:500; MilliporeSigma); mouse Nesprin1 MANNES1A(7A12) (#MA5-18077, 1:500; Thermo Fisher Scientific); mouse Nesprin3 (# MUB1317P,1:250; Nordic-MUBio); rabbit Emerin D3B9G XP (# 30853; CST); mouse MHC MF-20 (in house, Developmental Studies Hybridoma Bank at the University of Iowa). For Actin staining: Phalloidin Alexa Fluor-647(#A22287; 1:1,000; Thermo Fisher Scientific) was used. All Alexa Fluor dye-conjugated secondary antibodies against rabbit and mouse were obtained from Thermo Fisher Scientific and used at 1:2,000 dilution.

## Microscopy, image analysis, and quantitation

All samples were imaged on Leica SP8 scanning confocal microscope using 63X/1.4 NA or 20x/0.75 NA oil immersion objectives. All samples in a given experiment were imaged using the same laser intensities and gain settings without saturation. Multiple z-slices were acquired to image entire nuclei and quantitation was performed on maximum intensity projections of z-stacks in Fiji (ImageJ) (Schindelin et al, 2012). Nuclear fluorescence intensity measurements were performed in Fiji by generating a nuclear mask based on Hoechst DNA staining and applying the mask to other background-subtracted channels of the same image. This allowed us to measure the integrated densities for each detected nuclei across different channels in a multicolor imaging experiment. F-actin intensity measurements were performed in CellProfiler (McQuin et al, 2018) by determining the primary object as Hoechst-stained nuclei and the secondary object as Phalloidin-stained F-actin to segment out cell boundaries in each image. Integrated densities were measured in a Phalloidin channel for each segmented cell.

For Sun2C/N ratio quantification, Sun2C fluorescence intensities measured using the above method were divided by the corresponding Sun2N intensities per nucleus per condition generating a "Sun2C/N ratio." Ratios from different replicates were pooled together. For normalization across different conditions in a given experiment, all values were divided by the average value of the control group in that given experiment, thereby generating a "Norm Sun2C/N." "Norm Sun2C/GFP" ratios were generated in the same way by using GFP intensity in place of Sun2N. To quantify "Norm Int." for NE markers in different experiments, we used the integrated

densities per nuclei per condition for that particular NE marker across replicates and normalized it to the average value of the control group in that given experiment. To quantify percentage of cells with SUN2 signal in ER, we took images of cells either stained for endogenous SUN2 or expressing GFP-tagged constructs (SUN2 WT and mutants) in 2–3 different fields of view on two independent days. We manually counted SUN2/GFP-positive cells and determined whether SUN2/GFP signal was restricted to the nucleus or also detected in the ER for that particular cell. Percentage of ER-positive signal cells was calculated per field of view and represented as one data point. All calculations were done in MS Excel.

For wound healing assays, Brightfield images of 2–3 different wound regions per condition were acquired on Invitrogen EVOS FL system using 4x air objective. Each image was analyzed using Wound_healing_size_tool plugin(Suarez-Arnedo et al, 2020) in Fiji by adjusting the parameters to quantify the percentage cell-free region.

Densitometry analysis of Western blots was performed using the Analyze Gels function in Fiji. For oligomer gels, Sun2 oligomer (Sun2(O)) band intensity was divided by Sun2 monomer band intensity (Sun2(M)) for that particular sample/lane. Additionally, all other samples were adjusted to the control group/first lane. For Norm. GFP-SUN2 level quantitation, GFP band intensity was normalized to aTUB levels of the same sample. In addition, all samples were normalized to the average value of GFP-SUN2 WT. To calculate Sun2 relative abundance after CHX treatment, SUN2 (for WT) or GFP (GFP tagged SUN2 variants) band intensity was normalized to aTUB band intensity of that sample and then normalized to 0 h time point of that group/cell line. For Norm. GFP-SUN2 level quantitation, GFP band intensity from an untreated sample (0 h) CHX experiment was used and normalized to aTUB levels of the same sample.

All calculations were done in MS Excel. Prism GraphPad was used to perform statistical analysis and generate graphs. Figures were made in Adobe Illustrator. Schematic illustrations were made either in Adobe illustrator or created with BioRender.com.

## Wound healing assay

C2C12 cells were reverse transfected twice with siRNAs on Day 0 and Day 1 as described above. Cells were trypsinized on Day 1 evening and seeded in either one well of a 24 well plate (for live imaging) or ibidi chambers (fixed cell imaging) at a desirable density to obtain a confluent monolayer on the following day. On Day 2, cell monolayers were serum deprived by changing the media to DMEM containing 2% FBS and P/S. On Day 3, starved cell monolayers were scratched using a sterile 1 ml micropipette tip, washed twice, and maintained in a low serum condition. Wounded monolayers were imaged on Day 3 post-wound for the 0 h time point and Day 4 for the 24 h time point to assess the cell-free region. For IF staining in wounded monolayers, multiple ibidi chambers were prepare and fixed at 0, 6 or 8 h post wound.

## Protein extraction and Western blotting

C2C12 cells expressing different SUN2 mutants were transfected either with mCherry control or DN-KASH-expressing plasmids for 24 h and total cell lysates were resolved on reducing/nonreducing

gels and Western blots probed with GFP antibody to detect the mutant proteins.

For resolving oligomers on nonreducing gels, cell pellets were resuspended in 1x low SDS RIPA buffer (25 mM Tris HCl pH 7.6, 150 mM NaCl, 1% NP40 substitute, 1% sodium deoxycholate, 0.1% SDS) supplemented with 1x cOmplete Protease inhibitor cocktail, 20 mM N-Ethylmaleimide (#E3876; Sigma-Aldrich) and 250 U/ml Benzonase Nuclease (# E8263; Sigma) and kept on ice for 15 min. Lysates were gently passed through a 27-gauge syringe 10 times and left on ice for another 15 min. Cell lysates were cleared by centrifugation at 16,000*g* for 15 min at 4C. The obtained supernatant was divided into two equal parts and either combined with an equal volume of 2x sample buffer (125 mM Tris–HCl pH 6.8, 10% Glycerol, 0.01% Bromophenol Blue and 0.1% SDS) or with 2x sample buffer supplemented with 100 mM DTT. +/−DTT samples were then resolved on 6% SDS–PAGE without boiling the samples. Adapted from Lu et al (2008).

All other protein extracts were prepared by resuspending cell pellets in harsh lysis buffer (10 mM Tris–HCl pH 7.5, 150 mM NaCl, 0.2% NP-40, 0.25% sodium deoxycholate, 0.05% SDS, 1 mm EDTA) supplemented with 1x cOmplete Protease inhibitor cocktail and 250 U/ml Benzonase Nuclease(Franks et al, 2016). Cell lysates were passed through a 27-gauge syringe 10 times on ice and lysates were clarified by spinning at 14,000 rpm for 15 min at 4°C. The supernatant was transferred to a new tube, combined with SDS loading buffer, and heated at 95°C for 10 min. Equal amount of cell lysates were resolved on 10% SDS–PAGE.

For Western blotting, proteins were transferred to a nitrocellulose membrane using wet transfer. Membranes were blocked in TBST (0.25% Tween 20, 20 mM Tris, pH 8.0, and 137 mM NaCl) containing 5% non-fat milk for 30 min followed by incubation with primary antibodies overnight in a shaker at 4°C. Membranes were washed in TBST thrice; incubated with secondary antibodies for 1 h at RT, followed by three washes with TBST, and stored in PBS. Membranes were developed using ECL and the signal was detected using KwikQuant Imager (KindleBiosciences LLC) or Odyssey Imaging System (LI-COR Biosciences).

The following antibodies and corresponding dilutions were used: mouse anti-Sun2 (Sun2N) clone 3.1E (#MABT880,1:1,000; MilliporeSigma); rabbit anti-GFP (#ab290-50 *μ*l, 1:1,000; EnquireBio); rabbit anti-Torsin A/DYT1 (TOR1A) (#ab34540, 1:500; Abcam); mouse anti-alpha-Tubulin (aTUB) (#T5168, 1:4,000; Sigma-Aldrich). HRP-conjugated secondary antibodies against mouse proteins were obtained from Thermo Fisher Scientific and used at 1:5,000 dilution.

## Supplementary Information

## Acknowledgements

The authors would like to thank Prof. Daniel A. Starr (UC Davis) for helpful discussions, Jan Lammerding's lab (Cornell) for sharing plasmids. We would also like to thank all members of the Hetzer laboratory for helpful comments on the manuscript and a special thanks to Ukrae Cho and Kenneth Kuhn for their help with experiments. This work was supported by the NOMIS foundation (MW Hetzer), and the National Institutes of Health (R01 NS096786 to MW Hetzer).

## Author Contributions

R Sharma: data curation, formal analysis, and methodology.
MW Hetzer: conceptualization and funding acquisition.

## Conflict of Interest Statement

The authors declare that they have no conflict of interest.

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
