## [Reviewer comments · Life Science Alliance]

Disulfide bond in SUN2 regulate dynamic remodeling of LINC complexes at the nuclear envelope

Rahul Sharma and Martin Hetzer

DOI: <https://doi.org/10.26508/lsa.202302031>

Corresponding author(s): Martin Hetzer, Institute of Science and Technology Austria

Review Timeline:	Submission Date:	2023-03-07
	Editorial Decision:	2023-03-10
	Revision Received:	2023-05-02
	Accepted:	2023-05-02

Transaction Report:

Please note that the manuscript was previously reviewed at another journal and the reports were taken into account in the decision-making process at Life Science Alliance.

Reviewer #1 Review

Comments to the Authors (Required):

I stand by my original review. This manuscript describes and carefully characterizes an exciting tool. The revised manuscript satisfied my relatively minor concerns about the data. However, the revision does not address my major concern that this is a new tool, but that very little new is learned about the assembly or regulation of Sun-Nesprin interactions or LINC formation. I do not see a significant mechanistic advance for the LINC field here.

Reviewer #2 Review

Comments to the Authors (Required):

This ms is a revised version in which the authors have constructively addressed most of my concerns. The description and application of a conformation-specific Sun2 antibody is interesting for those invested in LINC complex biology and dynamics. While the data support the idea that conformational changes related to Sun2 oxidoreduction accompany differentiation, some limitations remain. For example, a C615A/S mutant construct would have helped to support the author's conclusions, especially since mutations the other partner in disulfide formation (C719) results in mislocalization and decreased protein stability. It is also not entirely clear why the authors chose to do resort to citing other references and providing indirect evidence for the KASH/Sun2 complex formation instead of simply probing for both partners directly, as is commonly done (and would have been easier in the opinion of this reviewer). That said, I don't feel that elaborate/additional experiments would significantly change the central conclusions that conformational changes/disulfide rearrangement in Sun2 accompany differentiation, or wound healing. Overall, the data mostly support the author's ideas however I have seen more definitive studies in this journal. Below I list two additional minor points for consideration by the authors:

1) Fig. S3- what causes this rather complex pattern of bands in the 75 kDa - 100 kDa range?

2) It would be useful to show/highlight the Cys sidechains in the structure depictions, it is not clear to the reader where the cys residues reside (Fig. S4A). Also, it would be best to use a homology model and use mouse, and not the human AA positions.

Reviewer #3 Review

Comments to the Authors (Required):

The authors dismissed my previous concerns regarding data interpretation without addressing these, stating that clear reasoning for these concerns was not provided. I have re-read my initial comments and these very clearly highlight my interpretation of the data shown in the manuscript and how this conflicts with the interpretation of the authors. This should be addressed as per my original review.

Also, the response to my second point about the cysteine mutant does not address the issue. I asked whether it had been shown that the C719A mutant folds correctly, and they answered that it has been shown that cysteine disulphide formation is not required for binding. These are not equivalent.

Reviewer #1 Review

Comments to the Authors (Required):

The gap in knowledge is that it is not understood how the oligomerization of LINC and interaction between SUN and KASH are regulated. This is an important question of broad interest to the cell biology community. The current available tools used for studying this in a cell are quite limited.

The major advance here is the characterization of a new tool to distinguish between bound and unbound SUN/KASH interactions, a commercially available antibody against the C-term of Sun2C. Figure 1 is the highlight of the manuscript. It demonstrates the utility of the Sun2C antibody. This reagent will be useful for studying biogenesis of LINC complexes and will be of high interest to specialists in the field working on the dynamics of LINC oligomerization. It is an important advance in that it is a new tool.

However, the rest of the manuscript is disappointing. The tool is well characterized, but in the end, very little new about LINC dynamics is learned. For example, the intriguing models proposed by the Nie et al structure are not sufficiently tested here. For the level of impact I would expect at this journal, I would want to see more of a biological, mechanistic advance. Furthermore, as described below, there are missing controls, poor statistics, and over interoperations of the data throughout the manuscript.

Major concerns

1. The characterization of the Sun2C antibody is the major advance in this paper. It would therefore be helpful to better show what the Sun2C antibody is binding on a structure. From Abcam that its raised against a peptide of the last 17 residues of Sun2C. It would be helpful to the reader if you showed a structure of the Sun2C/nesprin complex (like in Sosa et al 2012) and clearly marked the last 17 residues of Sun2C so we can see its relative position compared to the KASH peptide. It would also be helpful to mark the 3 cys in Sun2C discussed throughout the paper.
2. Furthermore, in Sosa et al, the KASH lid was shown to move much closer to the Sun2 cation loop in the apo Sun2C structure vs the KASH-bound structure. Is this the conformational change being detected by the Sun2C antibody? Perhaps some molecular dynamic modeling could show how the antigen is blocked in the absence of KASH. Since this is already clearly predicted by the apo structure, the significance and impact of the findings here are not high.
3. It took me a lot of digging to figure out it is a commercially available mAb. This should be clearly stated and described for the reader in the results. In the light of the work by Abcam, Figure 3B is kind of expected, but nice validation that the epitope is the same as the peptide used to make the antibody. It's a valuable tool, and the results should make it clear to others in the field how easy it is to get the Sun2C antibody.
4. A necessary control is missing for the dominant negative KASH experiment in Figure 1. Adding an Ala to the C-term of the KASH or deleting the C-terminal Pro residues would demonstrate it is acting as predicted.
5. Figure 2 and 4 native gel experiments have a reproducibility problem. They are missing error bars. Are the quantifications shown in D and E from a single blot? How many blots were performed?
6. The term Sun2 oligomers, used throughout, is quite confusing. I don't think it has been shown they are Sun2 oligomers. More needs to be done to determine what the ~200 kD band in the native gel westerns is. Is it SUN binding nesprins? A western or mass spec should identify the components of the larger complex. I would not predict the band to be Sun2 oligomers, as the three cys residues in the SUN domain should not be forming di-sulfide bonds with other Sun2 proteins. Sun2 trimers shouldn't stay together in a native gel. Some sort of in vitro biochemistry is probably needed to make any conclusions here. The Lu et al 2008 paper referenced was looking at the role of Sun1-Sun1 dimerization through a cys far away from the SUN domain. Does Sun2 have cys outside the SUN domain that may be complicating the analysis? Also, the smaller band in the DN-KASH lane suggests it's not Sun2 oligomerization, but rather Sun2/KASH binding.
7. In 3F, a big deal is made in the text that the C719F degrades faster. This might not have anything to do with KASH binding, it could just be if you mess with the cation loop of Sun2, the whole timing is unstable. This doesn't tell anything about SUN-KASH interactions. If C719F does degrade faster, that complicates the analysis of using this construct in later figures. It is likely just unstable, as the cation loop is probably important. This is a problem with the conclusions. In the C719 mutants throughout, is it that the Ab no longer recognizes the epitope, or (more likely) the protein degrades? Also, need to show a blot for this experiment.

8. The model in 3G is problematic. It is unlikely that C719 and C615 are not forming a di-sulfide bond when Sun2C is monomeric. Did the Nie et al structure see a di-sulfide bridge in the monomeric structure?
9. The statistics in Fig 5 need reworking. In A,C, there is no rescue of siSun2, and in D, everything is compared to siLuc instead of siSun2 + WT. Also, the text implies the siSun1/2 is worse, but there are no statistics supporting this claim. Also, the imaging is hard to see. Please show actin using a stronger objective so we can see the actin organization.
10. Conclusions, like in the middle of page 7, that at the wound edge there is a dynamic oxidation of Sun2 are too far and not supported. The C719A phenotype mimics the siSun2 unrescued cells. This could be a general knockdown of Sun2 and not due to specific loss of binding to KASH.

Minor comments

1. Please spell out myoblasts and myotubes. The MT acronym is especially difficult for cell biologists in this field who are used to it meaning microtubules.
2. The claim that this is the only way to distinguish LINC complex oligomerization is too strong. There is a biophysical tool for studying SUN oligomerization in live cells, FFS. FFS might not readily available for the average cell biologist, but at least should be mentioned and referenced.
3. In Figs 4,5, and 6, the results are normalized or compared to siLuc. But instead, they should be compared to the siSun2 rescued with WT Sun2.
4. Fig 7G lacks reproducibility. Fig 7 needs a siTor1a rescue experiment to validate the siRNA approach.
5. Page 6 claim that this is the first evidence for the conservation of C615 and C719 is too strong. The structures show why they are conserved, as they hold together a cation loop.
6. The model in Figure 8 suggests that Sun oligomers can move by the NPC into the INM. The literature suggests that they

Reviewer #2 Review

Comments to the Authors (Required):

In their manuscript titled "Disulfide bond in SUN2 regulate dynamic remodeling of LINC complexes at the nuclear envelope," the authors describe a conformation-specific antibody recognizing the C terminus of SUN2 (anti-SUN2C). Using this newly established reagent, the authors find that the C-terminal disulfide bond between C615 and C719 is important for SUN2 inner nuclear membrane (INM) localization, protein stability, and LINC complex assembly. The authors also show that while the highly conserved C577 is not involved in SUN2 INM localization, it influences SUN2 half-life and is required for SUN2 homo-oligomerization. Furthermore, C577 is required for the KASH-SUN2 interaction. While LINC complexes are relatively unperturbed upon C577 mutation compared to C719, both C577 and C719 are important for mediating F-actin polymerization. Lastly, the authors find that various mechanisms of ER stress cause more C-terminally reduced SUN2 to accumulate. Overall, this manuscript describes a highly useful, innovative set of reagents that can discriminate conformationally distinct states of Sun2 during diverse physiological processes such as migration and differentiation in situ. We recommend this manuscript for publication if the authors can address the following points:

Major points

1. The paragraph describing the data presented in Figure 1E is a bit confusing and should be reworded to more clearly state that Sun1 depletion makes more nesprins available to bind Sun2 and mask the epitope-hence less Sun2C staining. Also, it is unclear to the reader how depletion of Syne3 leads to more sun2C staining. Is Syne3 more dominantly expressed in myoblasts than Syne1? A clearer wording to better describe the rationale/conclusion would be helpful.
2. In Figure 2A and B, the authors cannot firmly conclude that the high MW species represent disulfide-linked Sun2 oligomers. Also, the antibodies used for blotting should be indicated below or besides the blot. Furthermore, it would be helpful to stain with anti-DNKASH Ab (or use a suitably epitope-tagged version) to confirm the identity of the band.
3. To this reviewer, it is not clear why overexpression of SUN1L would cause less SUN2 oligomerization in Figure 2B. In Figure 8, the authors clearly indicate a disulfide forming between C577 across SUN2 monomers. So even if the C719/C615 disulfide is reduced, shouldn't the C577 disulfide still be apparent?
4. The CHX chase experiment in Figure 3F is interesting but not entirely convincing. It would be helpful if the authors included the original blot(s) to support their interpretation. Furthermore, the black line on the graph in Figure 3F is not clearly labeled. What is this condition? To this reviewer, the GFP signal in Figure 3B looks almost higher for the C719A and C577A mutants compared to WT. Although Figure 3C quantifies this, the authors should add more values as n = 3 is somewhat low for this application.
5. In Figure 3, the authors introduce the domain architecture of SUN2 and highlight the C-terminal disulfide bond between C615 and C719. However, they only show data with the C719A mutation. It would be more complete to demonstrate the same consequence of making a C615A mutation using even one readout, i.e., that the construct is highly ER localized or leads to LINC complex mis-assembly.

6. In Figure 4 there are a few aspects of the system that are potentially confusing, i.e., the dominant-negative effect of SUN2-C719A on LINC assembly and compensation of endogenous SUN1 "masking" any phenotype from SUN2-C577A. It is also somewhat counterintuitive that C719A leads to improper LINC complex assembly despite interacting with KASH domains via C577. So perhaps to more clearly show that C577 is required for KASH interactions, the authors could perform a standard immunoprecipitation with WT, C719A, and C577A SUN2 and blot for SYNE3 or the DN-KASH construct. This point gets a little muddled in Figure 4 as is.

7. In this reviewer's opinion, Figure 5B and D are not sufficiently described in the text or figure legend. Figure 5B appears to be a rescue experiment and the authors should clearly state that the C719A and C577A mutants do not rescue the F-actin polymerization to the same degree as the WT SUN2. For example, the authors could report a percent rescue by these mutants compared to the WT. This would be more intuitive.

8. In Figure 7, the authors demonstrate that knocking down TorsinA causes a substantial increase in the amount of reduced/free SUN2. This is accompanied by a decrease in LINC complex function. The authors discuss this in the context of TorsinA depletion causing some degree of ER stress, however, they do not demonstrate any such effect. In many systems, depleting TorsinA does not cause detectable ER stress. The authors should either demonstrate that this condition causes ER stress or discuss/state alternative mechanisms by which TorsinA could contribute to the SUN2 redox state (not requiring additional experiments).

9. The immunoblot in Figure 7G should be repeated three times and the ratio of the SUN2(O)/(M) should be reported for each blot.

10. Please remove "dramatically" from the text. Also, this reviewer suggests to downscale the conclusions. "Demonstrates" or "clearly demonstrates" seem somewhat overused throughout the manuscript. More careful interpretation/phrasing (e.g/ consistent with, indicates, etc.) would be adequate. For example, "excessive binding ..." on page 5 is an overinterpretation. Based on the presented data, the authors cannot clearly separate disulfide bonding/redox state from structural changes or higher-order assembly influencing antibody binding (in all likelihood it might be a complex combination of these factors).

Minor points

1. The statement that non-reducing gels can only resolve disulfide-linked species is not correct.

2. There are many cases of improper comma use and small grammatical errors. The authors should go over the text with a fine-tooth comb to fix these.

3. The authors should more specifically discuss the functional consequences of accumulating reduced SUN2 upon ER stress. What would this imply for LINC complex integrity, differentiation, or cell migration?

4. Related to the above minor point, do the authors propose that the majority of the reduced SUN2 is produced by some degree of ER stress, or could this be an active process at the INM?

5. There are a few instances where the quantification for a figure panel is not the next panel within the figure. For example, the quantification for Figure 5A is panel C when it would be no change in the figure layout to simply call that panel B.

6. The results describing Figure 4A say loss of emerin at the NE is expected upon C719A expression. Why would this be expected?

7. Suggestion: it would be useful to show a basic ribbon diagram of the Sun2 structure with the cys sidechains highlighted.

8. Number and unit should be separated by one space throughout the manuscript.

Reviewer #3 Review

Comments to the Authors (Required):

The LINC complex bridges the nuclear envelope, connecting the nucleoplasm and cytoplasm, and fulfilling a number of roles in nuclear structure and function. It is formed of a complex between SUN and Nesprin proteins, which interact within the lumen. Here, the authors have identified an antibody that discriminates between different forms, corresponding to bound/unbound SUN molecules, which they use describe LINC dynamics within the nuclear envelope.

The data are of good quality and generally support conclusions. I have one major issue regarding the nature of the discriminating Sun2C Ab epitope that is fundamental to data interpretation in the manuscript, which must be addressed before it can be considered further.

My summary of the data regarding the targeting of the S2C Ab with regards to disulphide formation and Nesprin binding are as follows:

1. Sun2C Ab epitope is present in MBs and lost in MTs (when Nesprins become expressed) (1B,C)
2. Sun2C Ab epitope is lost upon Nesprin binding (1D,E)
3. Sun2 forms disulphides in MBs that are lost in MTs (2A)
4. Sun2 disulphides are lost upon binding to Nesprins (2B)

These findings indicate that Sun2C Ab recognises the disulphide form that is present in MBs prior to Nesprin-binding, and that the epitope is blocked upon loss of disulphide formation and interaction with Nesprin.

However, the authors interpret the data as suggesting that Sun2C Ab recognises the reduced form of the protein, and that Nesprin-binding induces disulphide formation. To me, this interpretation is incompatible with the data. The authors need to reconsider this and review their interpretation or prior further justification for their preferred model.

I would like to highlight that the interpretation I suggest is entirely compatible with our structural understanding of the SUN-KASH interaction. In absence of Nesprin, SUNs adopt an autoinhibited state in which the globular domains fail to trimerise, and in this context disulphide formation appears likely. Nesprin binding occurs very close to the disulphide bond, so it is consistent that the disulphide must be reduced to provide the conformational freedom necessary for the interaction. Further, I would suggest that the Sun2C Ab recognises a monomeric globular SUN domain in which the epitope is lost upon trimerization/complex formation.

I have a couple of additional related points:

- The authors use a C719A mutation to block disulphide formation and claim that this increases Ab binding and disrupts Nesprin binding, which fits with their model. However, has it been shown that C719A can form a SUN domain and interact with Nesprin in vitro? The residue is buried in an important location so alanine may be too drastic to support binding (serine would be a more conservative mutation in this case but would also benefit from biochemical validation). Hence, I am wondering whether the mutation renders the SUN domain monomeric, enabling Ab binding and blocking Nesprin binding?
- The authors show that a C-terminal deletion of Sun2 blocks Ab binding, and on this basis claim that the epitope must be within the last 17aa of the protein. However, this deletion is very unlikely to fold a SUN domain correctly, so it could be that the Ab recognises the folded structure, which is lost in this deletion.

March 10, 2023

RE: Life Science Alliance Manuscript #LSA-2023-02031-T

Dr. Martin Hetzer
Institute of Science and Technology Austria
Klosterneuburg 3400
Austria

Dear Dr. Hetzer,

Thank you for submitting your revised manuscript entitled "Disulfide bond in SUN2 regulate dynamic remodeling of LINC complexes at the nuclear envelope". We would be happy to publish your paper in Life Science Alliance pending final revisions necessary to meet our formatting guidelines.

- please address Reviewer 2's remaining minor point #2
- please address Reviewer 3's comment from the first review round regarding folding of the C719A mutant. This can be addressed as a possibility via added discussion.
- please upload your main manuscript text as an editable doc file
- please upload both your main and supplementary figures as single files
- please add a summary blurb/ alternate abstract and a category to our system
- please add the Twitter handle of your host institute/organization as well as your own or/and one of the authors in our system
- please use the [10 author names, et al.] format in your references (i.e. limit the author names to the first 10)
- please add the author contributions to the main manuscript text

Figure Check:

- in Figure 2C, please indicate the splices on the blot to remove the unnecessary lanes. This can be done by adding vertical lines where the lanes used to be, and indicating what the lines indicate in the figure legend.
- please add sizes next to the blots in Figure 3G
- please add a scale bar to Figure 5C and Figure 6C
- you may consider uploading Figure 8 as a graphical abstract, rather than as a figure. This is up to you.

A. FINAL FILES:

B. MANUSCRIPT ORGANIZATION AND FORMATTING:

Sincerely,

May 2, 2023

RE: Life Science Alliance Manuscript #LSA-2023-02031-TR

Dr. Martin Hetzer
Institute of Science and Technology Austria
Am Campus 1
Klosterneuburg, NO 3400
Austria

Dear Dr. Hetzer,

Thank you for submitting your Methods entitled "Disulfide bond in SUN2 regulate dynamic remodeling of LINC complexes at the nuclear envelope". It is a pleasure to let you know that your manuscript is now accepted for publication in Life Science Alliance. Congratulations on this interesting work.

DISTRIBUTION OF MATERIALS:

Again, congratulations on a very nice paper. I hope you found the review process to be constructive and are pleased with how the manuscript was handled editorially. We look forward to future exciting submissions from your lab.

Sincerely,
